# MICROADAM: Accurate Adaptive Optimization with Low Space Overhead and Provable Convergence

Ionut-Vlad Modoranu[1][*]     Mher Safaryan[1]     Grigory Malinovsky[2]     Eldar Kurtic[1]
Thomas Robert[1]     Peter Richtárik[2]     Dan Alistarh[1]

[1]Institute of Science and Technology Austria (ISTA)
[2]King Abdullah University of Science and Technology (KAUST)

## Abstract

We propose a new variant of the Adam optimizer [Kingma and Ba, 2014] called MICROADAM that specifically minimizes memory overheads, while maintaining theoretical convergence guarantees. We achieve this by compressing the gradient information before it is fed into the optimizer state, thereby reducing its memory footprint significantly. We control the resulting compression error via a novel instance of the classical *error feedback* mechanism from distributed optimization [Seide et al., 2014, Alistarh et al., 2018, Karimireddy et al., 2019] in which *the error correction information is itself compressed* to allow for practical memory gains. We prove that the resulting approach maintains theoretical convergence guarantees competitive to those of AMSGrad, while providing good practical performance. Specifically, we show that MICROADAM can be implemented efficiently on GPUs: on both million-scale (BERT) and billion-scale (LLaMA) models, MICROADAM provides practical convergence competitive to that of the uncompressed Adam baseline, with lower memory usage and similar running time. Our code is available at https://github.com/IST-DASLab/MicroAdam.

## 1 Introduction

The Adam [Kingma and Ba, 2014] adaptive optimizer and its variants [Reddi et al., 2019, Loshchilov and Hutter, 2019] has emerged as a dominant choice for training deep neural networks (DNNs), especially in the case of large language models (LLMs) with billions of parameters. Yet, its versatility comes with the drawback of substantial memory overheads: relative to naive SGD-based optimization, the Adam optimizer states doubles the memory overhead, as it requires storing two additional parameters for each variable. For large-scale models, these memory demands pose a significant challenge. In turn, this has spurred research into memory-efficient adaptive optimizers, such as AdaFactor [Shazeer and Stern, 2018], 8-bit Adam [Dettmers et al., 2021], or the very recent GaLore [Zhao et al., 2024] low-rank projection approach. Despite their popularity and practical utility, the above methods lack rigorous convergence guarantees, and often trade off memory reductions with decreased convergence in practice. This raises the question of whether it is possible to design adaptive optimizers that are not only memory-efficient, but also maintain strong theoretical and practical performance metrics.

**Contributions.** In this paper, we address this gap by introducing MICROADAM, an adaptive optimizer which guarantees low memory usage but also ensures provable convergence. *We develop our approach to improve the performance of finetuning LLMs* and mainly focus on the research question "are all gradient entries important for optimization?" To answer this question, we start from the idea that we can allow the (lossy) sparse projection of gradient information before it enters the optimizer states; crucially, different from prior work, we ensure convergence by correcting for the inherent error

---

[*]Correspondence to `ionut-vlad.modoranu@ista.ac.at`

38th Conference on Neural Information Processing Systems (NeurIPS 2024).

due to compression by employing a novel variant of *error correction*, a mechanism introduced for distributed optimization [Seide et al., 2014]. However, simply using error feedback would not lead to memory savings, since the size of the error correction buffer is comparable to the that of the original optimizer state. Instead, our main algorithmic innovation is in showing that the error feedback *can itself be compressed* in the context of adaptive optimization. This renders the memory overhead of error feedback negligible, while preserving convergence guarantees.

Specifically, on the theoretical side, we provide a new analysis showing that, under reasonable assumptions on the loss function being optimized and on the degree of compression, MICROADAM provably guarantees convergence, at asymptotically the same rate as AMSGrad [Zhou et al., 2024a], i.e. a version of Adam with general convergence guarantees, that fixes a fundamental technical issue in the Adam optimizer's proof [Reddi et al., 2019]. The key finding is that our approach allows for the overhead of compression to be shifted to the higher-order terms, where it should not impact practical convergence in common cases. This claim holds both for general smooth non-convex functions, and for non-convex functions under the Polyak-Lojasiewicz (PL) assumption, highlighting a trade-off between the degree of compression of the gradients, and that of the error feedback.

We complement our algorithmic and analytic results with an efficient GPU implementation of MICROADAM, which we validate for fine-tuning language models from the BERT [Devlin et al., 2018], OPT [Zhang et al., 2022] and LLaMA [Touvron et al., 2023] families, with hundreds of millions to billions of parameters. We show that, in practice, gradients can be projected to very high sparsity (99%), while the error correction can be stored at 4 bits per component, without loss of convergence. Concretely, our method can significantly improve upon the memory footprint of the extremely popular 8bit Adam [Dettmers et al., 2021] when fine-tuning models such as LLaMA2-7B/13B [Touvron et al., 2023], at similar or better accuracy. At the same time, MICROADAM provides better accuracy relative to high-compression heuristics such as GaLore [Zhao et al., 2024].

In summary, we provide a new theoretically-grounded approach to memory-efficient adaptive optimization, which has the advantage of providing both theoretical guarantees and good practical convergence, while being scalable to billion-parameter models. MICROADAM could therefore serve as a useful new tool for accurate and memory-efficient optimization of large models.

## 2   Related Work

We mainly focus on related work reducing the cost of optimizer states. Dettmers et al. [2021] considers this problem, specifically by performing fine-grained quantization of the optimizer states. Their work does not alter the Adam algorithm; instead, it deals with the challenge of accurately compressing the dynamically-changing meta-data sequence. As the name suggests, the space savings correspond to roughly halving the memory required by the optimizer states, relative to FP16. In the same vein, AdaFactor [Shazeer and Stern, 2018] and CAME [Luo et al., 2023] reduce memory cost by factorizing the second-order statistics, while the recent GaLore [Zhao et al., 2024] factorizes the gradients themselves before they enter the optimizer state (but does not use error correction). Importantly, these methods are *heuristics*: they do not provide theoretical guarantees under standard assumptions,[2] and in practice require careful tuning to preserve convergence [Luo et al., 2023]. By contrast, our method is theoretically justified, and provides good practical convergence. Earlier work by Anil et al. [2019] provides convergence guarantees for a compressed variant of Adagrad [Duchi et al., 2010] called SM3, improving upon earlier work by Spring et al. [2019]. However, it is not clear how to extend their approach to the popular Adam optimizer, and heuristic methods appear to provide superior performance [Luo et al., 2023].

Conceptually, our work is related to error feedback mechanisms studied in distributed optimization, e.g. [Seide et al., 2014, Alistarh et al., 2018, Karimireddy et al., 2019, Richtárik et al., 2021]. Specifically, Li et al. [2022] proved convergence of AdaGrad-like algorithms in conjunction with error feedback, in a distributed environment. Our focus is different: minimizing memory costs in the single-node setting: for this, *we show that the error correction buffer can itself be compressed.* We provide an analysis for the resulting new algorithm, and efficient CUDA implementations.

More broadly, scaling adaptive or second-order optimizers to large models is a very active area. Works such as GGT [Agarwal et al., 2019], Shampoo [Gupta et al., 2018] and M-FAC [Frantar et al.,

---

[2]GaLore [Zhao et al., 2024] does state convergence guarantees for a variant of the algorithm with fixed projections, but this is under a strong "stable rank" assumption, which may not hold in practice.

2021] provided quadratic-space algorithms that are still feasible to execute for moderate-sized DNNs, but will not scale for billion-parameter models. Follow-up work such as AdaHessian [Yao et al., 2020], Sophia [Liu et al., 2023], Sketchy [Feinberg et al., 2023] and EFCP [Modoranu et al., 2023], scaled these approaches via additional approximations. Of these, the closest work to ours is EFCP, which uses sparsification plus standard error feedback to compress the gradient window employed in the Fisher approximation of the Hessian. However, EFCP does not compress the error accumulator, assumes a different optimization algorithm (Natural Gradient [Amari, 2016]), lacks convergence guarantees, and does not scale to billion-parameter models.

## 3   The MICROADAM Algorithm

**Notation.** We consider a standard Adam-type algorithm, which we will augment for memory savings. We will use $f$ for the loss function, $d$ for the model size, $k$ for the gradient density (sparsity $d - k$), $\theta_t$ and $g_t$ for the model parameters and gradient at step $t$ respectively, $\eta_t$ for the learning rate, $\lambda$ for the weight decay parameter, $m_t$ and $v_t$ for the first and second moment of the gradient, $\epsilon$ for the numerical stability constant, $\beta_1$ and $\beta_2$ for the momentum coefficients for $m_t$ and $v_t$ respectively. Furthermore, we use $e_t$ for the error feedback (EF) vector, $b$ the number of bits for EF quantization, $m$ for the sliding window size, $\mathcal{G} = (\mathcal{I}, \mathcal{V})$ for the sliding window of size $m \times k$ that stores indices $\mathcal{I}$ and values $\mathcal{V}$ selected by the Top-K operator.

**Algorithm Description.** We provide pseudocode in Algorithm 1 and highlight the parts related to error feedback quantization in blue. The main idea is to compress the gradients via TopK sparsification before they enter the optimizer state, and to correct for the inherent error by applying error feedback $e_t \in \mathbb{R}^d$. *Instead of storing the optimizer state directly, we maintain a "sliding window" of highly-sparse past gradients and dynamically re-compute the Adam statistics at each step based on this window.* Yet, this alone does not improve space, as the error buffer partially negates the benefits of gradient compression. Instead, we prove that the error feedback accumulator can itself be compressed via quantization.

In detail, at step $t = 1$, the error feedback $e_1$ is completely zero, as initialized in line 2, and thus, at line 5 the accumulator $a_1$ will only contain the stochastic gradient $g_1$. At line 6, we perform the Top-K compression and only keep the top-1% of values $\mathcal{V}_1$ and their corresponding indices $\mathcal{I}_1$. The compression is equivalent to choosing the top-1% values in the left and right tails (outliers) due to the absolute value we apply on top of accumulator $a$. At line 7, we remove the outliers from the accumulator because they will be transferred to the buffer matrix $\mathcal{G}$. This step is equivalent to $e \leftarrow a - T_k(a)$ found in theoretical results. After line 7, what is left in $a$ is called the error feedback (e.g. the weights which were not chosen by Top-k). At line 8, we compute the statistics $\delta$ and $\Delta$ needed for quantization, and at line 9, we effectively quantize the accumulator (e.g. error feedback after line 7). At line 10 we update the buffer $\mathcal{G}$, in lines 11, 12 and 13 we compute the statistics $\hat{m}, \hat{v}$ (computed by squaring the entries of $\mathcal{G}$ element-wise) and update the model parameters.

For steps $t \geq 2$, the only change compared to $t = 1$ is that error feedback $e$ is not zero anymore. Since the error is compressed, we need to decompress it and add it to the gradient. This process happens at line 5 and it is the point where we feed back the error: the accumulator will store the gradient whose direction is corrected by the error (e.g. the cumulative history of weights not chosen by Top-k at the previous steps).

**Properties and Limitations.** We would like to point out that the resulting update $u_t = m_t/(\epsilon + \sqrt{v_t})$ will always be highly sparse when the window size $m$ is small. For illustration, if we use density $k = d/100$ (e.g. 1% density equivalent to 99% sparsity) with $m = 10$ and suppose that all rows in the indices matrix $\mathcal{I}$ are disjoint, then the overall density in the update $u_t$ will be 90%. The sparsity of $u_t$ increases if rows in $\mathcal{I}$ have common values. MICROADAM yields good results for LLM finetuning and pre-training computer vision models, as the experimental section shows. However, we noticed the update $u_t$ of MICROADAM is too sparse to be able to provide good enough updates for LLM pre-training. We believe this happens because the attention layers must receive dense updates to be able to learn the correlations between words.

**Dynamic Statistics.** In ADAMSTATS procedure in Algorithm 2 we implement the unrolled recursion of momentum $z_t \leftarrow \beta z_{t-1} + (1-\beta)g_t$ for the last $m$ sparse gradients as $z_t \leftarrow (1-\beta)\sum_{i=t-m}^{t} \beta^{t-i}g_i$ and we also perform the bias correction in the end. Because we compute $\hat{m}_t$ and $\hat{v}_t$ using the last $m$ sparse gradients in the window, in line 4 we dynamically determine the exponent $r$ for the decay factor $\beta^r$ based on the current optimization step $t$, $i^{th}$ row of the circular buffer $\mathcal{G}$ and the window size

$m$. The last gradient added to $\mathcal{G}$ will have $r = 0$, while the oldest gradient in $\mathcal{G}$ will have $r = m - 1$. In the end, we will add the values $\beta^r \mathcal{V}_i$ to the buffer $z$ at the corresponding indices $\mathcal{I}_i$, which is a fast operation because we only manipulate $1\%$ of values at a time. We discuss the efficient CUDA implementation in the Appendix.

**Algorithm Intuition.** To gain intuition, we illustrate the impact of compressing gradients via TopK before they are incorporated into the optimizer state for Adam, both with and without error feedback (EF). Figure 1 shows how EF fixes AdamW with Top-K compression. The plot on the left shows the optimization trajectory of the original Adam optimizer. The center plot illustrates the convergence of Top-K Adam when we only choose the largest coordinate from the accumulator (equivalent to $50\%$ sparsity since the problem is 2D). In the end, on the right side we show that adding EF to Top-K Adam recovers the same optimization trajectory as the original Adam optimizer. Extrapolating to higher dimensional problems, our MICROADAM approach helps recover the trajectory of the original Adam optimizer, while using less memory. The results clearly show that EF is essential for fast convergence. Besides, TopK with EF, which is a surrogate of MICROADAM, allows for competitive convergence relative to the uncompressed baseline. In Appendix F, we discuss the implications of Error Feedback applied to GaLore.

---

**Algorithm 1** Pseudocode for MICROADAM with quantized EF

1: **Input:** $\beta_1, \beta_2, \epsilon, \mathcal{G}, T, d, k$
2: $m_0, v_0 \leftarrow 0_d, 0_d$
   $\delta_1, \Delta_1 \leftarrow 0, 0$
   $e_1 \leftarrow 0_d^{4b}$
3: **for** $t = \{1, 2, ..., T\}$ **do**
4:      $g_t \leftarrow \widetilde{\nabla}_\theta f(\theta_t)$
5:      $a_t \leftarrow g_t + Q^{-1}(e_t, \delta_t, \Delta_t)$
6:      $\mathcal{I}_t, \mathcal{V}_t \leftarrow T_k(|a_t|)$
7:      $a_t[\mathcal{I}_t] \leftarrow 0$
8:      $\delta_{t+1}, \Delta_{t+1} \leftarrow min(a_t), max(a_t)$
9:      $e_{t+1} \leftarrow Q(a_t, \delta_{t+1}, \Delta_{t+1})$
10:     $\mathcal{G}_{i,:} \leftarrow (\mathcal{I}_t, \mathcal{V}_t)$
11:     $\hat{m}_t \leftarrow \text{ADAMSTATS}(\beta_1, \mathcal{G})$
12:     $\hat{v}_t \leftarrow \text{ADAMSTATS}(\beta_2, \mathcal{G}^2)$
13:     $\theta_{t+1} \leftarrow \theta_t - \eta_t \frac{\hat{m}_t}{\epsilon + \sqrt{\hat{v}_t}}$
14:     $i \leftarrow (i+1)\%m$
15: **end for**

---

**Algorithm 2** Adam Statistics, Quantization and Inverse Quantization

1: **procedure** ADAMSTATS($\beta, \mathcal{G}, t, m, d$)
2:      $z \leftarrow 0_d$
3:      **for** $i \in \{1, 2, ..., \min(t, m)\}$ **do**
4:         $r \leftarrow (t - i - 1)\%m$
5:         $z[\mathcal{I}_i] \leftarrow z[\mathcal{I}_i] + \beta^r \mathcal{V}_i$
6:      **end for**
7:      **return** $\frac{(1-\beta)z}{1-\beta^t}$
8: **end procedure**

1: **procedure** $Q(x, \delta, \Delta, b = 4)$
2:      $u \leftarrow \frac{\Delta - \delta}{2^b - 1}$
3:      $x_Q \leftarrow \lfloor \frac{x - \delta}{u} + \frac{1}{2} \rfloor$
4:      **return** $x_Q$
5: **end procedure**

1: **procedure** $Q^{-1}(x_Q, \delta, \Delta, b)$
2:      $u \leftarrow \frac{\Delta - \delta}{2^b - 1}$
3:      $x \leftarrow x_Q \cdot u + \delta$
4:      **return** $x$
5: **end procedure**

---

Figure 1: Optimization trajectories of Adam, TopK-Adam and TopK-Adam with EF applied on the Rosenbrock function $f(x, y) = (1 - x)^2 + 100(y - x^2)^2$ starting from $(x_0, y_0) = (-\frac{1}{2}, 1)$. Notice the extremely "jagged" profile of TopK-Adam without EF, and the recovered convergence when EF is added.

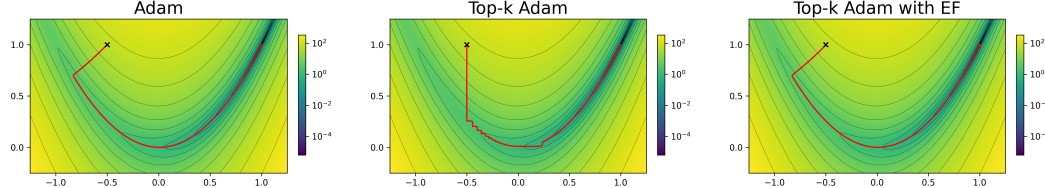

## 3.1 Efficient Implementation

A direct implementation (e.g., in Pytorch) of the previous algorithm would not bring significant benefits, and would in fact might slow down optimization in terms of wall-clock time. To realize the theoretical gains, we detail a GPU-aware implementation below.

**Accumulator** $a_t$. First, we do not use an additional accumulator tensor $a_t$; instead, we use a CUDA kernel to dequantize the error buffer, and store the result in the *grad* attribute of the model parameters. This allows us to accumulate the error feedback into the gradients, without allocating a full or half precision $d$-dimensional array. Each component of the EF has 4 bits and the entire EF is stored in an array of size $d/2$ of *uint8* values.

**Top-K.** Since we run on LLMs with billions of parameters, naive storage of the sparse indices would require using an *int64* type for the indices matrix $\mathcal{I}$, assuming that the Top-K operator is applied globally to all the parameters in $a_t$. To avoid this cost, we apply Top-K in blocks of fixed size $B_d < 2^{15} - 1 = 32767$ and store block-relative indices in *int16* format (during the development of MICROADAM, PyTorch did not have support for *uint16*). Applying Top-K per row to $a_t$ reshaped to 2D is not only faster, but provides the block-relative indices directly.

Computing Top-K in blocks also allows us to allocate and efficiently use CUDA shared memory blocks to dynamically compute the statistics $\hat{m}_t$ and $\hat{v}_t$ for Adam, as described in the ADAMSTATS procedure in Algorithm 2. We allocate the maximum possible shared memory for each thread block and store $\hat{m}_t$ (first half) and $\hat{v}_t$ (second half) at consecutive locations in the shared memory. Once these statistics are computed, it is easy to update the model parameters. Note that the block-relative indices returned by Top-K will be directly used as indices in the shared memory array of CUDA thread blocks to retrieve values from $\mathcal{I}$ and $\mathcal{V}$.

**Quantization metadata.** Our approach also stores two additional vectors $\delta$ and $\Delta$ used for quantization. Since the quantization block size $B_q$ is set to a very large value (e.g. $100K$), the space required for these two arrays becomes negligible in comparison to the buffer $\mathcal{G}$ and error feedback $e$.

**Practical memory usage.** We note that we apply MICROADAM per layer, and that the size of quantization statistics $\delta$ and $\Delta$ are allocated based on the layer size. Having many such small tensors may result in slightly sub-optimal memory allocation from Pytorch. This is why our reported memory usage can be higher than the theoretical usage for small models, in the 100M parameter range; these effects disappear for billion-parameter models, where the savings are significant.

### 3.2 Memory footprint analysis for the optimizer states and comparison with other methods

We now compare the theoretical memory footprint of MICROADAM with AdamW [Loshchilov and Hutter, 2019], AdamW-8 bits [Dettmers et al., 2021], and GaLore [Zhao et al., 2024], *focusing on memory usage of the optimizer states $m_t$ and $v_t$, each of size $d$, expressed in bytes* (B). For concreteness, we report the practical memory usage for the optimizer state for a Llama-2 7B model for each optimizer.

- **AdamW** stores states in *float32* format (4 B per component), resulting in a total memory footprint of $4d + 4d = 8d$ (B), while using *bfloat16* would result in $4d$ (B) memory. We will refer to these memory footprints as $M_{AW32} = 8d$ (B) $= 50.21$ (GB) and $M_{AW16} = 4d$ (B) $= 25.10$ (GB).

- **AdamW-8 bit** stores states in 8-bit format (1 B per component), both with $d$ components, with memory footprint of $M_{AW8} = d + d = 2d$ (B) $= 12.55$ (GB).

- **MICROADAM** stores the error feedback $e$ in 4-bit format (0.5 B per component) and the sliding window $\mathcal{G}$ that stores the indices $\mathcal{I}$ in *int16* and $\mathcal{V}$ in *bfloat16* format. Both have $m \times k$ components, each requiring 2 B per component. In the end, for $m = 10$ and $k = d/100$, the memory footprint is $M_{\mu A}(m) = 0.5d + 4mk$ (B) $= 0.9d$ (B) $= 5.65$ (GB), that is, half of AdamW-8bit.

- **GaLore.** Given a neural network with $L$ layers, where each layer has weight matrix $W_i \in \mathbb{R}^{A_i \times B_i}$ (shaped as a 2D matrix), the model size $d = \sum_{i=1}^{L} A_i B_i$. GaLore uses a rank-$r$ compression via the SVD composition as $W_i = USV^T$, where $U \in \mathbb{R}^{A_i \times A_i}$ and we choose the first $r$ columns of $U$ as $R_i \in \mathbb{R}^{A_i \times r}$ to project the gradient of $W_i$ to the $r$-dimensional space. As a consequence, the dimension $d$ shrinks to $d_r = \sum_{i=1}^{L} A_i r = r \sum_{i=1}^{L} A_i$, which represents the total number of components to be stored in the GPU memory only for the projection matrices $R_i$. If we suppose they are stored in *bfloat16* (2 B per component), then the entire memory usage for low-rank projection would be $2d_r$. Note that some layers in the model are rank-1 and they do not need compression, but will still have associated states in Adam, which means they must be tan into consideration when computing the theoretical memory (we will use $\epsilon_1$ for memory of rank-1 layers). In addition, we have to store the states $m_t$ and $v_t$ for AdamW in *bfloat16* format, which adds another $4d_r$ bytes. In sum, the total memory footprint of GaLore is $M_{GLAW16}(r) = 6d_r + 2\epsilon_1$ (B), while for the 8-bit version we get $M_{GLAW8}(r) = 4d_r + 2\epsilon_1$ (B). In the end, the practical

memory usage for Llama-2 7B is $M_{GLAW8}(256) = 1.36$ (GB), $M_{GLAW8}(1024) = 5.43$ (GB), $M_{GLAW16}(256) = 2.04$ (GB) and $M_{GLAW8}(1024) = 8.15$ (GB).

**Discussion.** Assume our goal is to obtain a lower memory footprint compared to AdamW-8 bit. We fix the gradient density to $k = d/100$ and we have to determine the number of gradients (window size) $m$ for MICROADAM in order to be competitive with AdamW-8 bit.

For this, we have to solve the equation $M_{\mu A}(m) = M_{AW8}$ for $m$, resulting in $m_{\max} = 37.5$. Specifically, if we use a gradient history of $m < m_{\max}$ gradients in $\mathcal{G}$, MICROADAM will have theoretical memory savings. We will see that, in practice, this history size $m = 10$ is more than enough for good practical results. As entries in the window past this range are dampened extremely significantly, their influence is negligible. In Appendix D, we provide Python code to compute the theoretical memory usage for the three optimizers for Llama-2 7B.

## 4 Convergence Guarantees for MICROADAM

In this section, we present our theoretical framework. We begin by introducing and discussing the analytical assumptions used in our theory, providing an analytical view of the proposed MICROADAM algorithm, along with two theoretical convergence results.

### 4.1 Gradient and Error Compression

We now define two classes of compression operators widely used in the literature.

**Assumption 1.** *The gradient compressor $\mathcal{C} : \mathbb{R}^d \to \mathbb{R}^d$ is $q$-contractive with $0 \le q < 1$, i.e.,*

$$\|\mathcal{C}(x) - x\| \le q \|x\|, \quad \text{for any } x \in \mathbb{R}^d.$$

The compression we use in Algorithm 1 is the TopK compressor $T_k$ selecting top $k$ coordinates in absolute value. This is known to be contractive with $q = \sqrt{1 - k/d}$. Another popular contractive compressor is the optimal low-rank projection of gradient shaped as a $d \times d$ matrix, in which case $q = \sqrt{1 - R/d}$ where $R$ is the projection rank.

The second class of compressors, which we use for the error feedback, requires unbiasedness and relaxes the constant in the uniform bound.

**Assumption 2.** *The error compressor $\mathcal{Q} : \mathbb{R}^d \to \mathbb{R}^d$ is unbiased and $\omega$-bounded with $\omega \ge 0$, namely,*

$$\mathbb{E}[\mathcal{Q}(x)] = x, \quad \|\mathcal{Q}(x) - x\| \le \omega \|x\|, \quad \text{for any } x \in \mathbb{R}^d.$$

One example of $\omega$-bounded compressor, a version of which is used in Algorithm 2, is the randomized rounding quantizer we employ, whose properties we provide below.

**Lemma 1.** *Consider Algorithm 2 with randomized rounding, i.e., for a vector $x \in \mathbb{R}^d$ with $\delta = \min_i x_i$ and $\Delta = \max_i x_i$, let $\hat{x}_i := \lfloor \frac{x_i - \delta}{u} + \xi \rfloor u + \delta$ be the $i$-th coordinate of the quantized vector $\hat{x}$, where $\xi \sim U[0,1]$ is the uniform random variable and $u = \frac{\Delta - \delta}{2^b - 1}$ is the quantization level. Then*

$$\mathbb{E}[\hat{x}] = x, \quad \|\hat{x} - x\| \le \frac{\sqrt{d-2}}{2^b - 1} \frac{\Delta - \delta}{\sqrt{\Delta^2 + \delta^2}} \|x\|, \quad \text{for all } x \in \mathbb{R}^d.$$

Next, we provide an "analytical" view of our method in Algorithm 3. Essentially, we use the contractive compressor $\mathcal{C}$ for compressing the error corrected gradient information $g_t + e_t$, and the unbiased compressor $\mathcal{Q}$ to compress the remaining compression error $g_t + e_t - \mathcal{C}(g_t + e_t)$.

---

**Algorithm 3** MICROADAM: Analytical View

---

1: Input: parameters $\beta_1$, $\beta_2 \in (0,1)$, $\epsilon > 0$, step-size $\eta > 0$, $\theta_1 \in \mathbb{R}^d$, $e_1 = m_0 = v_0 = \hat{v}_0 = 0_d$
2: **for** $t = \{1, 2, ..., T\}$ **do**
3:      $g_t = \widetilde{\nabla}_\theta f(\theta_t)$             $\diamond$ Compute unbiased stochastic gradient
4:      $\tilde{g}_t = \mathcal{C}(g_t + e_t)$             $\diamond$ Add accumulated error $e_t$ and compress
5:      $e_{t+1} = \mathcal{Q}(e_t + g_t - \tilde{g}_t)$          $\diamond$ Update and compress the error
6:      $m_t = \beta_1 m_{t-1} + (1 - \beta_1)\tilde{g}_t$      $\diamond$ Update first-order gradient moment
7:      $v_t = \beta_2 v_{t-1} + (1 - \beta_2)\tilde{g}_t^2$      $\diamond$ Update second-order gradient moment
8:      $\hat{v}_t = \max(v_t, \hat{v}_{t-1})$            $\diamond$ Apply AMSGrad normalization
9:      $\theta_{t+1} = \theta_t - \eta \frac{m_t}{\sqrt{\hat{v}_t} + \epsilon}$      $\diamond$ Update the model parameters
10: **end for**

---

It is clear from this description that our objective with these two compressors, $\mathcal{C}$ and $\mathcal{Q}$, is to approximate the dense gradient information $g_t + e_t$ using two compressed vectors: $\tilde{g}_t = \mathcal{C}(g_t + e_t)$ and $\mathcal{Q}(g_t + e_t - \tilde{g}_t)$. However, in doing so, we inevitably lose some information about $g_t + e_t$ depending on the degree of compression applied to each term. Thus, the condition $(1 + \omega)q < 1$ required by our analysis can be seen as preventing excessive loss of information due to compression.

## 4.2 Convergence Guarantees for General Smooth Non-convex Functions

Next, we state our algorithm's convergence guarantees under standard assumptions, stated below:

**Assumption 3** (Lower bound and smoothness). *The loss function $f \colon \mathbb{R}^d \to \mathbb{R}$ is lower bounded by some $f^* \in \mathbb{R}$ and $L$-smooth, i.e., $\|\nabla f(\theta) - \nabla f(\theta')\| \leq L \|\theta - \theta'\|$, for any $\theta, \theta' \in \mathbb{R}^d$.*

**Assumption 4** (Unbiased and bounded stochastic gradient). *For all iterates $t \geq 1$, the stochastic gradient $g_t$ is unbiased and uniformly bounded by a constant $G \geq 0$, i.e., $\mathbb{E}[g_t] = \nabla f(\theta_t)$, $\|g_t\| \leq G$.*

**Assumption 5** (Bounded variance). *For all iterates $t \geq 1$, the variance of the stochastic gradient $g_t$ is uniformly bounded by some constant $\sigma^2 \geq 0$, i.e., $\mathbb{E}[\|g_t - \nabla f(\theta_t)\|^2] \leq \sigma^2$.*

**Main Result.** The above assumptions are standard in the literature, e.g. [Défossez et al., 2022, Li et al., 2022, Xie et al., 2023, Zhou et al., 2024a]. Under these conditions, if the two compressors satisfy the basic condition $(1 + \omega)q < 1$, we show:

**Theorem 1. (Non-convex convergence rate)** *Let Assumptions 1, 2, 3, 4, 5 hold and $q_\omega := (1+\omega)q < 1$. Then, choosing $\eta = \min\{\frac{\epsilon}{4LC_0}, \frac{1}{\sqrt{T}}\}$, MICROADAM (Algorithm 3) satisfies*

$$\frac{1}{T}\sum_{t=1}^{T} \mathbb{E}[\|\nabla f(\theta_t)\|^2] \leq 2C_0 \left( \frac{f(\theta_1) - f^*}{\sqrt{T}} + \frac{L(\sigma^2 + C_2^2 G^2)}{\epsilon\sqrt{T}} \right) + \mathcal{O}\left( \frac{G^3(G+d)}{T} \right)$$

*with constants $C_0 := \sqrt{\frac{4(1+q_\omega^2)^3}{(1-q_\omega^2)^2} G^2 + \epsilon}$ and $C_2 := \omega q(1 + \frac{2q_\omega}{1-q_\omega^2})$.*

**Discussion.** First, notice that the leading term $\frac{1}{\sqrt{T}}$ of the rate is the optimal convergence speed for non-convex stochastic gradient methods [Ghadimi and Lan, 2016]. Furthermore, the obtained convergence rate $\mathcal{O}(\frac{1}{\sqrt{T}} + \frac{d}{T})$ asymptotically matches the rate of uncompressed AMSGrad in the stochastic non-convex setup [Zhou et al., 2024a]. Hence, the added compression framework of the MICROADAM together with error feedback mechanism can slow down the convergence speed only up to some constants including the dimension. Evidently, the additional constants $C_0$ and $C_2$ affected by compression and appearing in the leading terms can be easily estimated once the compressors are fixed. Besides, if we store the full error information without applying $\mathcal{Q}$ compressor (i.e., $\omega = 0$, $q_\omega = q$), then MICROADAM reduces to the single-node Comp-AMS method by Li et al. [2022] recovering the same convergence rate. The full proof is provided in the Appendix.

## 4.3 Convergence Rate for Non-Convex Functions under the PL Condition

Next, we show that we can obtain even stronger bounds when the objective satisfies the PL condition:

**Assumption 6** (PL-condition). *For some $\mu > 0$ the loss $f$ satisfies Polyak-Lojasiewicz (PL) inequality*

$$\|\nabla f(\theta)\|^2 \geq 2\mu(f(\theta) - f^*), \quad \text{for any } \theta \in \mathbb{R}^d.$$

In this case, we can show:

**Theorem 2. (PL convergence rate)** *Let Assumptions 1, 2, 3, 4, 5 and 6 hold, and $q_\omega < 1$. Then, choosing $\eta = \min\{\frac{\epsilon}{4LC_0}, \frac{2C_0 \log T}{\mu T}\}$, MICROADAM (Algorithm 3) satisfies*

$$\mathbb{E}[f(\theta_{T+1})] - f^* \leq \frac{2 \log T}{T} \left( \frac{LC_0^2}{\mu} \frac{\sigma^2 + (C_1 + C_2^2)G^2}{\mu\epsilon} + \frac{C_0(1+C_1)(1+d)G^2}{\mu\sqrt{\epsilon}} \right) + \widetilde{\mathcal{O}}\left( \frac{G^4(G+d)}{T^2} \right)$$

*with constant $C_1 := \frac{\beta_1}{1-\beta_1}(1 + C_2) + \frac{2q_\omega}{1-q_\omega^2}$.*

**Discussion.** In contrast to the general non-convex setup, the study of non-convex analysis under the PL condition for AMSGrad or Adam-type methods is much less extensive. The only work we found analyzing the PL condition, which claims to be the first in this direction, focuses on Adam when $\beta_2 \to 1$, achieving a convergence rate of $\mathcal{O}(\frac{1}{T})$ [He et al., 2023]. However, our MICROADAM is based on AMSGrad normalization, and no constraint on $\beta_2$ is imposed in the analysis. Therefore, similar to

the general non-convex case, we are able to achieve the best-known convergence rate in the leading term, up to a logarithmic factor. The third, higher-order term has higher constant dependencies, but they should be negligible as the term is dampened by $T^2$. Hence, in this case as well, the theory predicts that the convergence rate of the algorithm should be similar to the uncompressed version, modulo a constant that can be controlled using the compression parameters.

## 5 Experiments

We now validate our optimizer experimentally. We focus on comparing MICROADAM with Adam, Adam-8bit, GaLore and CAME in the context of LLM finetuning on different tasks and with SGD, Adam and AdamW-8bit in the context of ResNets on ImageNet. Concretely, we test our optimizer in full finetuning (FFT) scenario on BERT-Base/Large [Devlin et al., 2018] and OPT-1.3B [Zhang et al., 2022] on GLUE/MNLI and Llama2-7B/13B [Touvron et al., 2023] on the GSM8k math reasoning dataset and on the Open-Platypus instruction tuning dataset, as well as pre-training ResNet models on ImageNet. We provide full details regarding training settings hyper-parameters in Appendix B.

**Finetuning results on GLUE/MNLI.** We first test our integration of MICROADAM in HuggingFace Transformers [Wolf et al., 2020] on moderate-sized language models such as BERT-Base/Large (110M and 335M parameters) and OPT-1.3B, comparing with Adam, Adam-8bit, CAME and GaLore. The results are shown in Table 1. Certain optimizers, notably CAME and GaLore, had numerical stability issues across runs; for a fair comparison, we report the numbers for the run with maximum accuracy. We emphasize that all methods were tuned using the same protocol.

The results show that MICROADAM achieves comparable memory usage to the state-of-the-art heuristics Adam-8bit and GaLore, while being surprisingly lower than CAME on all tasks. The memory savings for GaLore are more visible when the model size increases, which follows our analysis of theoretical memory usage. However, we see that these gains come at a significant *accuracy cost* for GaLore: across all tasks, it drops at least 1% accuracy relative to MICROADAM. For BERT-Base we ran GaLore with a higher SVD re-computation frequency $T = 20$ ($10\times$ lower) and the results did not improve, but its running time was much higher. Relative to 8bit Adam, MICROADAM uses essentially the same memory, but achieves slightly better accuracy.

From these results, we conclude that MICROADAM can provide better accuracy relative to other memory-efficient methods on moderate-sized models, at similar space costs. We show training loss curves in Appendix C.

Table 1: Finetuning results on GLUE/MNLI. We report the entire memory usage read from the GPU during training, that includes the optimizer state, activations and gradients. The asterisk flags the runs for which one or two seeds did not converge (we report the run with maximum performance).

| Model | Metric | MICROADAM ($m = 10$) | Adam | Adam-8b | CAME | GaLore $r = 256$ |
|---|---|---|---|---|---|---|
| BASE (110M) | train loss | 0.2651 | 0.4228 | 0.3402 | 0.6431* | 0.3908* |
| | accuracy | 85.10% | 83.53% | 84.61% | 76.13%* | 83.82%* |
| | memory | 2.55 GB | 2.70 GB | 2.53 GB | 2.69 GB | 2.53 GB |
| LARGE (335M) | train loss | 0.2509 | 0.3857 | 0.2876 | 0.6658* | 0.3768* |
| | accuracy | 86.17% | 84.79% | 86.18% | 75.23%* | 84.90%* |
| | memory | 5.98 GB | 6.64 GB | 6.04 GB | 6.59 GB | 5.85 GB |
| OPT-1.3B (1.3B) | train loss | 0.2122 | 0.2066 | 0.2611 | 0.4959 | 0.2831 |
| | accuracy | 88.18% | 87.90% | 87.81% | 83.15% | 87.70 |
| | memory | 15.28 GB | 17.66 GB | 15.00 GB | 17.13 GB | 13.66 GB |

**Finetuning results for LLaMA2 on GSM-8k.** Next, we perform finetuning on Llama-2 7B/13B on GSM-8k, a challenging grade-school-level mathematical reasoning dataset. The baseline model obtains extremely low zero-shot accuracy on this task and therefore fine-tuning is necessary. In this setup, we compare MICROADAM with Adam and Adam-8bit in terms of evaluation accuracy and memory usage. In Table 2 we show our results for 3 training epochs, global batch size 32 with micro-batch (per-device) size 1, max sequence length 512 on a single GPU, which are the standard

parameters for this task. We integrated our optimizer with the `llm-foundry` repository of MosaicML and tested via `lm-evaluation-harness`.

For the 7B model, out results show that MICROADAM can allow accurate full fine-tuning of a 7B model on this task using a single 40GB GPU. Moreover, MICROADAM preserves accuracy relative to Adam, with lower memory usage than the well-optimized implementation of 8bit AdamW, and marginally lower running time for the shorter gradient window $m = 10$. Increasing the window size $m$ to 20 gradients leads to slightly better accuracy, at the cost of higher runtime and space, but still in the 40GB limit. Running GaLore in this setup was infeasible since using SVD decomposition for all layers in the model was too slow. Preliminary experiments (with high runtimes) did not yield competitive accuracy. We show training loss curves in Appendix C.

The results show that MICROADAM allows for full accuracy recovery on this task as well relative to Adam, despite using 50% less memory. (The memory usage and runtime are very similar to those in Table 2 and are therefore omitted from Table 3.) Moreover, MICROADAM obtains consistently better accuracy relative to Adam-8b, especially on the more challenging ARC-c task.

Table 2: FFT results for Llama-2 7B/13B on GSM-8k.

| LLaMA-2 size | Optimizer | Accuracy | State | Total | Runtime |
|---|---|---|---|---|---|
| 7B | **Adam** | 34.50% | 25.1 GB | 55.2 GB | 1h 17m |
| | **Adam-8b** | 34.34% | 12.55 GB | 42.5 GB | 1h 18m |
| | **MICROADAM** ($m = 10$) | 34.72% | 5.65 GB | 37.1 GB | 1h 8m |
| | **MICROADAM** ($m = 20$) | 35.10% | 8.25 GB | 39.7 GB | 1h 37m |
| 13B | **Adam** | 47.08% | 48.42 GB | >80 GB | 1h 20m |
| | **Adam-8b** | 45.19% | 24.21 GB | >80 GB | 1h 17m |
| | **MICROADAM** ($m = 10$) | 44.88 % | 10.9 GB | 70 GB | 1h 38m |

**Finetuning results for LLaMA2-7B on Open-Platypus.** Finally, in Table 3 we present FFT results with various optimizers on the popular instruction-tuning Open-Platypus dataset [Lee et al., 2023]. To ensure fair comparisons, we perform the same grid search for each optimizer to find the best performing learning-rate, while keeping all other hyperparameters at their default values. We use $m = 10$ gradients for the sliding window and gradient density $k = 1\%$. Evaluations are conducted following the standard few-shot setup of the Open LLM Leaderboard [Beeching et al., 2023] on the following datasets: ARC-c [Clark et al., 2018], HellaSwag [Zellers et al., 2019], MMLU [Hendrycks et al., 2021], and Winogrande [Sakaguchi et al., 2019].

Table 3: FFT results on instruction-following Open-Platypus [Lee et al., 2023] dataset. The results show that MICROADAM fully recovers accuracy relative to baseline Adam, and outperforms the 8bit variant, despite using less memory.

| Optimizer | Memory | Average Accuracy | ARC-c 25-shot | HellaSwag 10-shot | MMLU 5-shot | Winogrande 5-shot |
|---|---|---|---|---|---|---|
| AdamW | 67.17 GB | 62.10 | 52.56 | 77.38 | 45.53 | 72.93 |
| Adam-8b | 53.93 GB | 61.84 | 51.96 | 77.51 | 44.11 | 73.79 |
| MICROADAM | 46.63 GB | 62.36 | 53.07 | 77.46 | 45.04 | 73.87 |

**Pre-training results for ResNets on ImageNet.** In Table 4 we present our results for pre-training (from scratch, randomly initialized weights) for ResNet-18/50 on ImageNet (see Figure 6 and Figure 7 in Appendix C). We compare our MICROADAM with SGD, Adam and AdamW and report the training loss, validation accuracy and only the optimizer state memory because the total memory usage is not an issue for ResNets on ImageNet (here we focus on the results to emphasize the pre-training performance). For ResNet-18, AdamW reaches the lowest training loss, followed by AdamW-8bit, MICROADAM and in the end MICROADAM. However, despite slightly larger loss, MICROADAM yields the best result among all optimizers, with 2% more than the highly tuned SGD, while having the lowest memory footprint for the optimizer states. For ResNet-50, AdamW-8bit reaches the lowest training loss, followed by AdamW, MICROADAM and SGD. The validation accuracy for AdamW and AdamW-8bit is surprisingly small compared to SGD and MICROADAM. As it is widely known

in the community, Adam variants have lower performance than SGD for Computer Vision tasks and MICROADAM fixes this issue (see the Discussion section for an intuitive explanation for this phenomenon).

Table 4: Pre-training results for ResNet-18 and ResNet-50 on ImageNet.

| Model | Metric | SGD | AdamW | AdamW-8bit | MICROADAM |
|-------|--------|-----|-------|------------|-----------|
| ResNet-18 | Train Loss | 1.416 | 1.087 | 1.104 | 1.218 |
|  | Accuracy | 69.96% | 69.83% | 70.13% | 71.86% |
|  | State Size | 44.59 MB | 89.18 MB | 22.30 MB | 10.03 MB |
| ResNet-50 | Train Loss | 0.9770 | 0.5344 | 0.5158 | 0.7732 |
|  | Accuracy | 76.24% | 72.05% | 72.48% | 77.37% |
|  | State Size | 97.49 MB | 194.98 MB | 48.75 MB | 21.94 MB |

**Pre-training LLMs.** In this section we explain why we do not include LLM pre-training results. Our motivation is twofold. First, MICROADAM is mainly designed for low-memory finetuning, and the experimental section shows that MICROADAM achieved this goal. Surprisingly, updating only 10% of the weights at each step yields to significantly better performance compared to SGD for ResNets on ImageNet. Secondly, our experiments on LLM pre-training showed difficulties in achieving the same performance compared to AdamW-8bit. Our explanation is that projection matrices from the attention layers must receive dense updates to learn the correlations between words. In contrast to the convolutional filters for CV models, the weights in attention are much larger and try to capture global correlations (features) between words, while the convolutional filters are smaller and capture local features.

**Discussion.** In Appendix A we provide information about the optimization set, intuitive explanations for the implicit regularization effect of MICROADAM, as well as an overview of our results.

## 6    Limitations and Broader Impact

The MICROADAM algorithm we propose is designed and tested with fine-tuning workloads in mind, where the user aims to minimize the memory cost of optimizing over a powerful pre-trained model. Additional work is needed to adapt our approach to the case of LLM pre-training, which presents a different set of challenges, both in terms of implementation and optimization trajectory. We plan to undertake this study in future work as the current implementation works for ResNets.

Another limitation we aim to address in future work is that we have only focused on sparsity as a form of gradient projection. However, our theoretical analysis also applies to low-rank projection of gradients. We believe that our practical implementation can be extended to this case as well, although providing a general, accurate, and efficient implementation will require non-trivial efforts.

Our work introduces a new, accurate, and memory-efficient optimizer for fine-tuning LLMs. The major positive impact of our approach is its ability to maintain performance while reducing memory requirements, thereby lowering the cost of running experiments due to the reduced hardware expenses. It is important to note that while our optimizer can enhance performance and reduce costs, we do not have control over the neural network applications trained with it.

## Acknowledgements

The authors thank Razvan Pascanu, Mahdi Nikdan and Soroush Tabesh for their valuable feedback, the IT department from Institute of Science and Technology Austria for the hardware support and Weights and Biases for the infrastructure to track all our experiments. Mher Safaryan has received funding from the European Union's Horizon 2020 research and innovation program under the Marie Sklodowska-Curie grant agreement No 101034413.

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

# Contents

# A   Additional Explanations and Experimental Details

**Optimization set.** The parameters included in the optimization set usually vary depending on the model and optimizer type. For GLUE, we do not include the embeddings in the optimization set for any of the optimizers because our experiments showed no significant difference when optimizing the embeddings. Moreover, it is more fair for GaLore which would have an increased memory usage due to the projection matrix for the embeddings. For LLaMa2 and ResNet models, we include all layers in the optimization set, regardless of their type. This means that MICROADAM updates at most 10% of the weights in each layer. In the original work, GaLore was applied only to a subset of layers, such as Q-, K-, V-, O-, up-, down- and gate-projection layers. Applying GaLore to all layers in the same way as we do with MICROADAM would result in much larger memory usage because of the projection matrices for the embeddings. Moreover, computing SVD for the embedding layer would be infeasible. As a result, we omit the GaLore for LLaMa2 and ResNet experiments.

**Implicit regularization.** In our results so far, we observed MICROADAM having better performance for a few LLM finetuning tasks, but especially for the ResNet/ImageNet results, where the difference was statistically significant (around 1%). Our intuition for this behavior mainly comes from the accuracy curves in Figure 6 and Figure 7. The trajectories of MICROADAM and SGD are typical for regularized training. We hypothesize that MICROADAM has an implicit regularization mechanism because the model update $u_t$ is 90% sparse, which leads to only updating 10% of the model parameters at each step in each layer. In contrast to a 100% dense update $u_t$, a sparse update would not change the model parameters as much as a dense one. In the ResNet experiments, all optimizers used the same regularization parameter $\lambda = 1e - 4$, but the accuracy graph shows that MICROADAM is more regularized than SGD, while the graphs for AdamW and AdamW-8bit look like the regularization does not have any effect.

**Discussion.** In summary, the experimental results have shown that MICROADAM can recover the state-of-the-art accuracy of the the uncompressed Adam baseline, while providing significant memory gains and matching wall-clock speed on billion-parameter models. Specifically, our approach matches and outperforms Adam-8b and CAME both in terms of memory use and in terms of final accuracy. Relative to the high-compression GaLore method, MICROADAM provides consistently higher accuracy, as well as more stable practical convergence. We conclude that MICROADAM should be a good alternative to Adam-8bit in memory-constrained settings, and that the empirical results appear to validate our theoretical predictions.

# B   Training Settings and Hyper-parameters

In this section we provide details about the hyper-parameters that we used for each model and dataset. We train all our models in *bfloat16* format, tune the learning rates on a grid and report the best accuracy among 3 seeds (7, 42 and 1234) and report the results for the best configuration that converged.

All Adam variants use default parameters $\beta_1 = 0.9, \beta_2 = 0.999, \epsilon = 10^{-8}$ and the regularization parameter $\lambda$ is 0 for finetuning and $3e - 4$ for ImageNet pre-training. MICROADAM uses a window size of $m = 10$ gradients with $k = 1\%$ density (equivalent to 99% sparsity and quantization bucket size is set to 64 for the error feedback.

For GaLore we use rank $r = 256$ and the SVD update interval is set to $T = 200$, as suggested by the original paper. We run our experiments on NVidia GPUs A100-SXM4-80GB, H100-80GB and on RTX 3090 with 24GB RAM in single GPU setup.

## B.1   GLUE/MNLI

For GLUE/MNLI, we used the learning rate grid $\{1e - 6, 3e - 6, 5e - 6, 7e - 6, 1e - 5, 3e - 5, 5e - 5, 7e - 5\}$ for all optimizers and models. Certain optimizers diverge for specific seeds. Next, we provide some details about hyper-parameters for each optimizer individually.

**MICROADAM.** We use $m = 10$ gradients in the sliding window, $k = 1\%$ density (e.g. 99% sparsity) and quantization bucket size 64 (we also tried 100 000, but this didn't affect performance or memory usage in a meaningful way).

**Adam and Adam-8bit.** All hyper-parameters mentioned above apply for these two main baseline optimizers.

**GaLore.** We use rank $r = 256$ and SVD update interval $T \in \{20, 200\}$. In the original GaLore paper, the authors tune both learning rate and in our experiments we keep scale fixed to value 1 and augment the learning rate grid with the values $\{1e-4, 3e-4, 5e-4, 7e-4\}$.

**CAME.** This optimizer has some additional parameters that we keep to default values, such as $\beta_3 = 0.9999$. Instead of $\epsilon$, it uses $\epsilon_1 = 1e - 30$ and $\epsilon_2 = 1e - 16$. The authors mention that the learning rate should be much smaller than Adam's and because of that we augment the learning rate grid with the values $\{1e-7, 3e-7, 5e-7, 7e-7\}$.

### B.2 GSM-8k.

For GSM-8k, we used the learning rate grid $\{1e-5, 2e-5, 3e-5, 4e-5, 5e-5, 6e-5, 7e-5, 8e-5, 9e-5\}$ and reported the model with the best evaluation accuracy. We found that different versions for `PyTorch`, `lm-eval-harness` and `llm-foundry` have large variance in the results.

**MICROADAM.** We use similar settings as for GLUE/MNLI above in terms of other hyper-parameters.

### B.3 ImageNet

For ImageNet, we integrate our MICROADAM in the FFCV repository, which is highly tuned for ResNets and SGD. We use $E = 100$ epochs, batch size 1024, cosine learning rate schedule with warmup and image resolution $224 \times 224$ and precision *bfloat16*. We started from the initial learning rate $\eta = 1.024$ tuned in the repository which scored highest accuracy for SGD. This learning rate also worked well for MICROADAM, but it didn't work for AdamW and AdamW-8bit. For these two Adam variants we divided the learning rate by 2 until the models converged.

**ResNet-18.** For AdamW, the learning rate is $\eta = 0.016$ and for AdamW-8bit is $\eta = 0.032$.

**ResNet-50.** For AdamW, the learning rate is $\eta = 0.008$ and for AdamW-8bit is $\eta = 0.008$.

## C Training Graphs

In this section we show training loss curves for BERT-Base, BERT-Large and OPT-1.3b on GLUE/MNLI and Llama-2 7B/13B on GSM-8k and ResNet-18/50 on ImageNet.

Figure 2: Training curves for BERT-Base on GLUE/MNLI

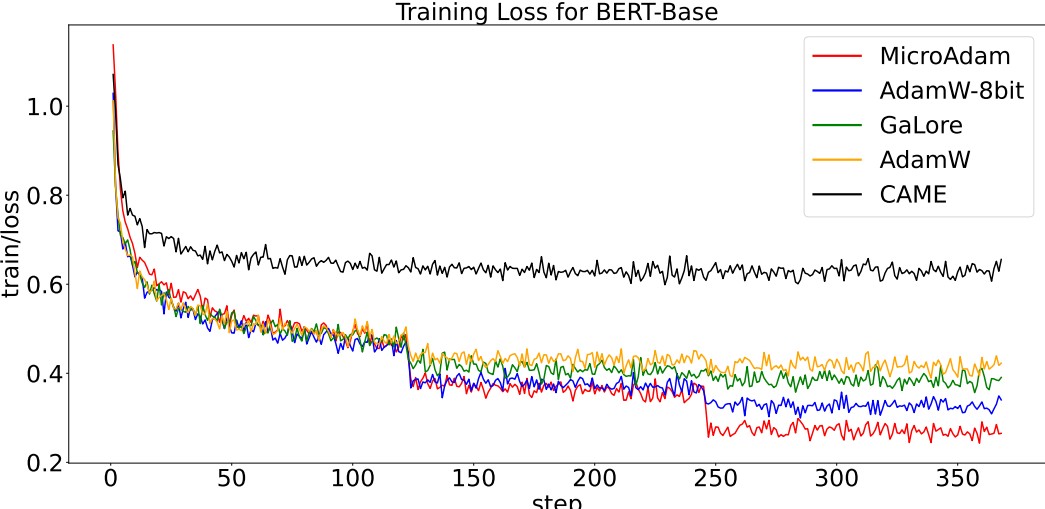

Figure 3: Training curves for BERT-Large on GLUE/MNLI

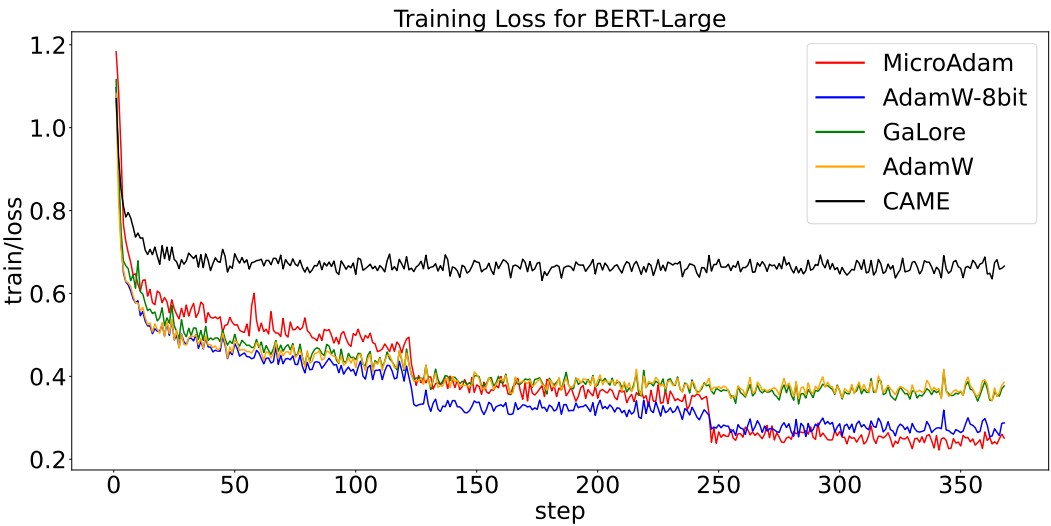

Figure 4: Training curves for OPT-1.3B on GLUE/MNLI

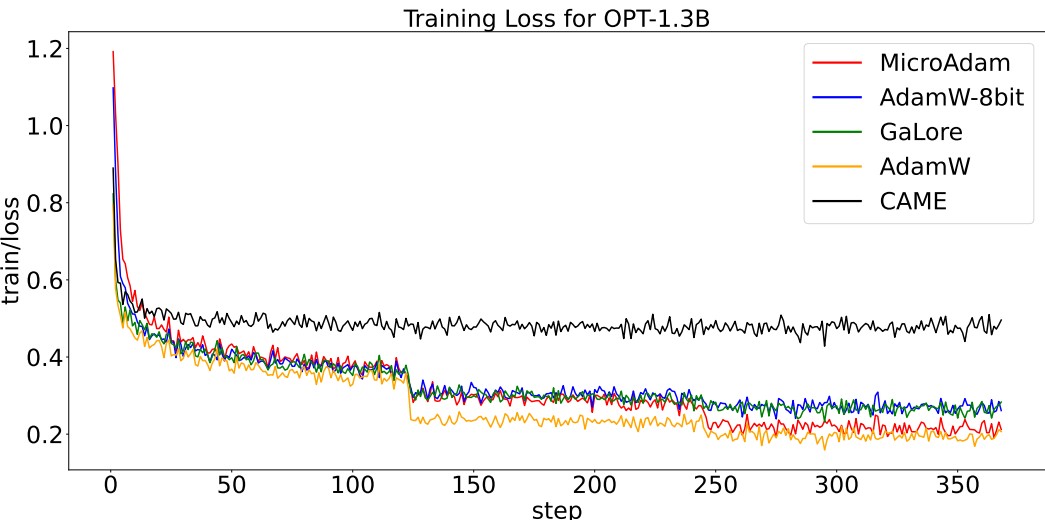

## D   Memory footprint for the optimizer state

In this section we provide a python script to simulate the memory usage for our optimizer's state for Llama2-7b model. Note that the theoretical memory usage will always be slightly lower than the actual allocated memory on the GPU because PyTorch usually allocates more. To run this script, run the following commands:

```python
import math

d = 6_738_415_616 # actual number of parameters for Llama-2 7b
k = math.ceil(d / 100)
m = 10

M_AW32 = 8 * d / (2 ** 30)
M_AW16 = 4 * d / (2 ** 30)
M_AW8 = 2 * d / (2 ** 30)
M_muA = (0.5 * d + 4 * m * k) / (2 ** 30)    # B to GB
```

Figure 5: Training curves for Llama-2 7B on GSM-8k

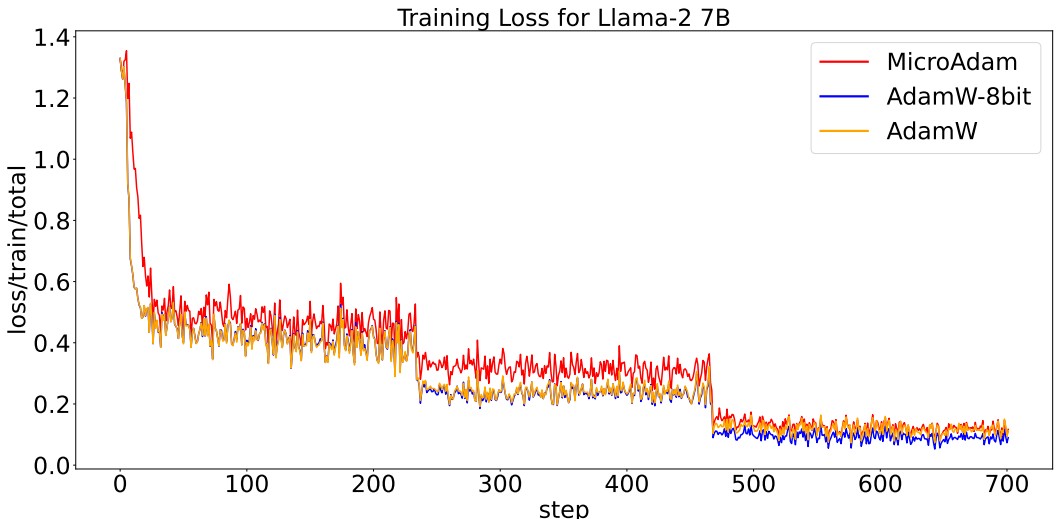

Figure 6: Pre-training for ResNet-18 on ImageNet

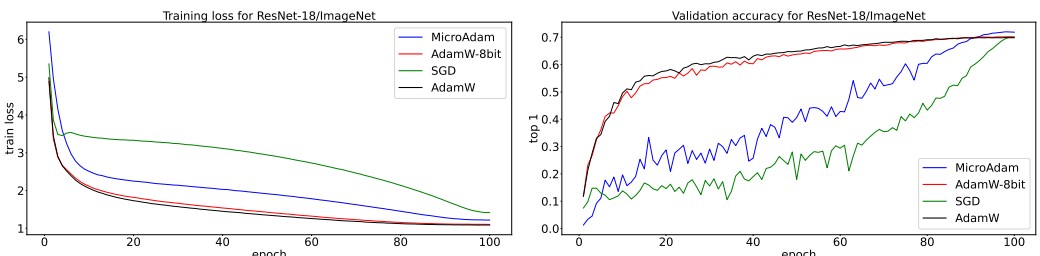

Figure 7: Pre-training for ResNet-50 on ImageNet

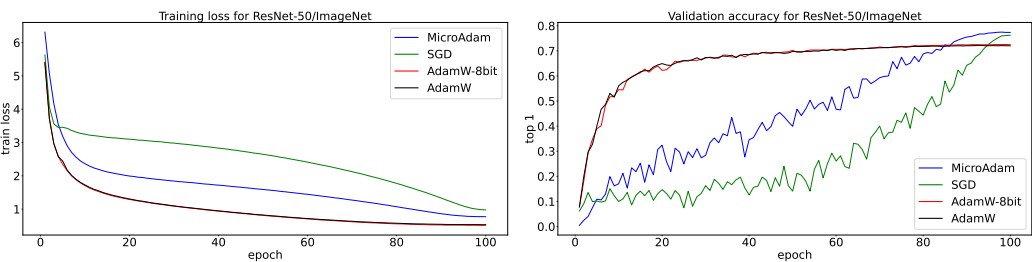

```python
print(f'{M_AW32=:.2f}_GB')
print(f'{M_AW16=:.2f}_GB')
print(f'{M_AW8=:.2f}_GB')
print(f'{M_muA=:.2f}_GB')

GL_sumA = 1_423_872 # sum_Ai from Llama-2 7B
epsilon = 266_240 # sum of sizes for rank-1 layers

for bits, const in [(8, 4), (16, 6)]:
    for rank in [256, 1024]:
        dr = rank * GL_sumA
        M_GLAW_rank = (const * dr + 2 * epsilon) / (2 ** 30)
        print(f'M_GLAW{bits}_rank{rank}={M_GLAW_rank:.2f}_GB')
```

# E Deferred Proofs

At time step $t$, let the uncompressed stochastic gradient be $g_t = \widetilde{\nabla}_\theta f(\theta_t)$, the error accumulator be $e_t$, and the compressed gradient after the error correction be $\tilde{g}_t = \mathcal{C}(g_t + e_t)$. The second moment computed by the compressed gradients is denoted as $v_t = \beta_2 v_{t-1} + (1 - \beta_2)\tilde{g}_t^2$, and $\hat{v}_t = \max\{\hat{v}_{t-1}, v_t\}$ is the AMSGrad normalization for the second-order momentum. Besides the first-order gradient momentum $m_t$ used in the algorithm description, we define similar running average sequence $m'_t$ based on the uncompressed gradients $g_t$.

$$m_t = \beta_1 m_{t-1} + (1 - \beta_1)\tilde{g}_t \quad \text{and} \quad m'_t = \beta_1 m'_{t-1} + (1 - \beta_1)g_t,$$

Note that $m'_t$ is used only in the analysis, we do not need to store or compute it. By construction we have

$$m_t = (1 - \beta_1)\sum_{\tau=1}^{t}\beta_1^{t-\tau}\tilde{g}_\tau, \quad m'_t = (1 - \beta_1)\sum_{\tau=1}^{t}\beta_1^{t-\tau}g_\tau$$

Denote by $\zeta_t = e_{t+1} - (e_t + g_t - \tilde{g}_t) = \mathcal{Q}(e_t + g_t - \tilde{g}_t) - (e_t + g_t - \tilde{g}_t)$ the compression noise from $\mathcal{Q}$. Due to unbiasedness of the compressor $\mathcal{Q}$ (see Assumption 2), we have $\mathbb{E}[\zeta_t \mid \theta_t, g_t, \tilde{g}_t, e_t] = 0$. Also, from the update rule of $e_{t+1}$ we get $e_{t+1} = e_t + g_t - \tilde{g}_t + \zeta_t$. Moreover, we use the following auxiliary sequences,

$$\mathcal{E}_{t+1} := \beta_1 \mathcal{E}_t + (1 - \beta_1)e_{t+1} = (1 - \beta_1)\sum_{\tau=1}^{t+1}\beta_1^{t+1-\tau}e_\tau.$$

$$\mathcal{Z}_{t+1} := \beta_1 \mathcal{Z}_t + (1 - \beta_1)\zeta_{t+1} = (1 - \beta_1)\sum_{\tau=1}^{t+1}\beta_1^{t+1-\tau}\zeta_\tau.$$

## E.1 Intermediate Lemmas

**Lemma 1.** *Consider Algorithm 2 with randomized rounding, i.e., for a vector $x \in \mathbb{R}^d$ with $\delta = \min_i x_i$ and $\Delta = \max_i x_i$, let $\hat{x}_i := \lfloor \frac{x_i - \delta}{u} + \xi \rfloor u + \delta$ be the $i$-th coordinate of the quantized vector $\hat{x}$, where $\xi \sim U[0, 1]$ is the uniform random variable and $u = \frac{\Delta - \delta}{2^b - 1}$ is the quantization level. Then*

$$\mathbb{E}[\hat{x}] = x, \quad \|\hat{x} - x\| \leq \frac{\sqrt{d-2}}{2^b - 1}\frac{\Delta - \delta}{\sqrt{\Delta^2 + \delta^2}}\|x\|, \quad \text{for all } x \in \mathbb{R}^d.$$

*Proof.* The unbiasedness can be verified directly from the definition for each coordinate. Without loss of generality assume that $\delta = x_1 \leq x_2 \leq \cdots \leq x_{d-1} \leq x_d = \Delta$. By construction of the quantization, we have $|\hat{x}_1 - x_1| = |\hat{x}_d - x_d| = 0$ and $|\hat{x}_i - x_i| \leq u$ for the remaining coordinates $2 \leq i \leq d - 1$. Then

$$\|\hat{x} - x\|^2 = \sum_{i=1}^{d}|\hat{x}_i - x_i|^2 \leq (d-2)u^2 \leq \frac{(d-2)u^2}{\Delta^2 + \delta^2}\|x\|^2,$$

which completes the proof. $\qquad\square$

**Lemma 2.** *Under Assumptions 1-5, for all iterates $t$ and $T$ we have*

$$\|m'_t\| \leq G, \quad \text{and} \quad \sum_{t=1}^{T}\mathbb{E}[\|m'_t\|^2] \leq T\sigma^2 + \sum_{t=1}^{T}\mathbb{E}[\|\nabla f(\theta_t)\|^2].$$

*Proof.* The first part follows from triangle inequality and the Assumption 4 on bounded stochastic gradient:

$$\|m'_t\| = (1 - \beta_1)\left\|\sum_{\tau=1}^{t}\beta_1^{t-\tau}g_\tau\right\| \leq (1 - \beta_1)\sum_{\tau=1}^{t}\beta_1^{t-\tau}\|g_\tau\| \leq G.$$

For the second claim, the expected squared norm of average stochastic gradient can be bounded by

$$\mathbb{E}\left[\|g_t\|^2\right] = \mathbb{E}\left[\|g_t - \nabla f(\theta_t))\|^2\right] + \mathbb{E}[\|\nabla f(\theta_t)\|^2] \leq \sigma^2 + \mathbb{E}[\|\nabla f(\theta_t)\|^2], \tag{1}$$

where we use Assumption 5 that $g_t$ is unbiased with bounded variance. Let $g_{t,j}$ denote the $j$-th coordinate of $g_t$. Applying Jensen's inequality for the squared norm, we get

$$
\begin{aligned}
\mathbb{E}[\|m_t'\|^2] &= \mathbb{E}\left[\left\|(1-\beta_1)\sum_{\tau=1}^{t}\beta_1^{t-\tau}g_\tau\right\|^2\right] \\
&\leq (1-\beta_1)\sum_{\tau=1}^{t}\beta_1^{t-\tau}\mathbb{E}[\|g_\tau\|^2] \\
&\leq \sigma^2 + (1-\beta_1)\sum_{\tau=1}^{t}\beta_1^{t-\tau}\mathbb{E}[\|\nabla f(\theta_\tau)\|^2],
\end{aligned}
$$

Summing over $t = 1, \ldots, T$, we obtain

$$
\sum_{t=1}^{T}\mathbb{E}[\|m_t'\|^2] \leq T\sigma^2 + (1-\beta_1)\sum_{t=1}^{T}\sum_{\tau=1}^{t}\beta_1^{t-\tau}\mathbb{E}[\|\nabla f(\theta_\tau)\|^2] \leq T\sigma^2 + \sum_{t=1}^{T}\mathbb{E}[\|\nabla f(\theta_t)\|^2],
$$

which completes the proof. $\qquad\square$

**Lemma 3.** *Let $q_\omega = (1+\omega)q < 1$. Under Assumptions 1-5, for all iterates $t$ we have*

$$
\|e_t\|^2 \leq \frac{4q_\omega^2}{(1-q_\omega^2)^2}G^2,
$$

$$
\mathbb{E}[\|e_{t+1}\|^2] \leq \frac{4q_\omega^2}{(1-q_\omega^2)^2}\sigma^2 + \frac{2q_\omega^2}{1-q_\omega^2}\sum_{\tau=1}^{t}\left(\frac{1+q_\omega^2}{2}\right)^{t-\tau}\mathbb{E}[\|\nabla f(\theta_\tau)\|^2].
$$

*Proof.* We start by using Assumption 1, 2 on compression and Young's inequality to get

$$
\begin{aligned}
\|e_{t+1}\|^2 &= \|\mathcal{Q}(g_t + e_t - \mathcal{C}(g_t + e_t))\|^2 \\
&\leq (1+\omega)^2 q^2 \|g_t + e_t\|^2 \\
&\leq q_\omega^2(1+\rho)\|e_t\|^2 + q_\omega^2\left(1+\frac{1}{\rho}\right)\|g_t\|^2 \\
&\leq \frac{1+q_\omega^2}{2}\|e_t\|^2 + \frac{2q_\omega^2}{1-q_\omega^2}\|g_t\|^2, \qquad (2)
\end{aligned}
$$

where (2) is derived by choosing $\rho = \frac{1-q_\omega^2}{2q_\omega^2}$ and the fact that $q_\omega < 1$. For the first claim we recursively apply the obtained inequality and use bounded gradient Assumption 4. For the second claim, initialization $e_1 = 0$ and the obtained recursion imply

$$
\begin{aligned}
\mathbb{E}[\|e_{t+1}\|^2] &\leq \frac{2q_\omega^2}{1-q_\omega^2}\sum_{\tau=1}^{t}\left(\frac{1+q_\omega^2}{2}\right)^{t-\tau}\mathbb{E}[\|g_\tau\|^2] \\
&\overset{(1)}{\leq} \frac{4q_\omega^2}{(1-q_\omega^2)^2}\sigma^2 + \frac{2q_\omega^2}{1-q_\omega^2}\sum_{\tau=1}^{t}\left(\frac{1+q_\omega^2}{2}\right)^{t-\tau}\mathbb{E}[\|\nabla f(\theta_\tau)\|^2],
\end{aligned}
$$

which concludes the lemma. $\qquad\square$

**Lemma 4.** *Let $q_\omega = (1+\omega)q < 1$. Under Assumptions 1-5, for all iterates $t$ we have*

$$
\|\zeta_t\| \leq \omega q\left(1+\frac{2q_\omega}{1-q_\omega^2}\right)G, \quad \text{and} \quad \|\mathcal{Z}_t\| \leq \omega q\left(1+\frac{2q_\omega}{1-q_\omega^2}\right)G.
$$

*Proof.* Using the bounds defining compressors and Lemma 3, we get

$$
\begin{aligned}
\|\zeta_t\| &= \|\mathcal{Q}(e_t + g_t - \tilde{g}_t) - (e_t + g_t - \tilde{g}_t)\| \\
&\leq \omega\|e_t + g_t - \tilde{g}_t\| = \omega\|e_t + g_t - \mathcal{C}(e_t + g_t)\| \\
&\leq \omega q\|e_t + g_t\| \\
&\leq \omega q\|e_t\| + \omega q\|g_t\| \\
&\leq \omega q\left(1+\frac{2q_\omega}{1-q_\omega^2}\right)G.
\end{aligned}
$$

For the second claim, recall the definition of $\mathcal{Z}_t$ and apply triangle inequality:

$$\|\mathcal{Z}_t\| \leq (1 - \beta_1) \sum_{\tau=1}^{t} \beta^{t-\tau} \|\zeta_\tau\| \leq \omega q \left(1 + \frac{2q_\omega}{1 - q_\omega^2}\right) G.$$

$\square$

**Lemma 5.** *For the moving average error sequence $\mathcal{E}_t$, it holds that*

$$\sum_{t=1}^{T} \mathbb{E}[\|\mathcal{E}_t\|^2] \leq \frac{4Tq_\omega^2}{(1 - q_\omega^2)^2} \sigma^2 + \frac{4q_\omega^2}{(1 - q_\omega^2)^2} \sum_{t=1}^{T} \mathbb{E}[\|\nabla f(\theta_t)\|^2].$$

*Proof.* Let $e_{t,j}$ be the $j$-th coordinate of $e_t$ and denote

$$K_t := \sum_{\tau=1}^{t} \left(\tfrac{1+q_\omega^2}{2}\right)^{t-\tau} \mathbb{E}[\|\nabla f(\theta_\tau)\|^2].$$

Applying Jensen's inequality and Lemma 3, we get

$$\mathbb{E}[\|\mathcal{E}_t\|^2] = \mathbb{E}\left[\left\|(1 - \beta_1)\sum_{\tau=1}^{t} \beta_1^{t-\tau} e_\tau\right\|^2\right]$$

$$\leq (1 - \beta_1) \sum_{\tau=1}^{t} \beta_1^{t-\tau} \mathbb{E}[\|e_\tau\|^2]$$

$$\leq \frac{4q_\omega^2}{(1 - q_\omega^2)^2} \sigma^2 + \frac{2q_\omega^2(1 - \beta_1)}{(1 - q_\omega^2)} \sum_{\tau=1}^{t} \beta_1^{t-\tau} K_\tau,$$

Summing over $t = 1, \ldots, T$ and using the technique of geometric series summation leads to

$$\sum_{t=1}^{T} \mathbb{E}[\|\mathcal{E}_t\|^2] \leq \frac{4Tq_\omega^2}{(1 - q_\omega^2)^2} \sigma^2 + \frac{2q_\omega^2(1 - \beta_1)}{(1 - q_\omega^2)} \sum_{t=1}^{T} \sum_{\tau=1}^{t} \beta_1^{t-\tau} K_\tau$$

$$\leq \frac{4Tq_\omega^2}{(1 - q_\omega^2)^2} \sigma^2 + \frac{2q_\omega^2}{(1 - q_\omega^2)} \sum_{t=1}^{T} K_t$$

$$= \frac{4Tq_\omega^2}{(1 - q_\omega^2)^2} \sigma^2 + \frac{2q_\omega^2}{(1 - q_\omega^2)} \sum_{t=1}^{T} \sum_{\tau=1}^{t} \left(\frac{1 + q_\omega^2}{2}\right)^{t-\tau} \mathbb{E}[\|\nabla f(\theta_\tau)\|^2]$$

$$\leq \frac{4Tq_\omega^2}{(1 - q_\omega^2)^2} \sigma^2 + \frac{4q_\omega^2}{(1 - q_\omega^2)^2} \sum_{t=1}^{T} \mathbb{E}[\|\nabla f(\theta_t)\|^2],$$

The desired result is obtained. $\square$

**Lemma 6.** *Let $q_\omega = (1 + \omega)q < 1$. Under Assumptions 1-5, for all iterates $t \in [T]$ and coordinates $i \in [d]$, the following bound holds*

$$\hat{v}_{t,i} \leq \frac{4(1 + q_\omega^2)^3}{(1 - q_\omega^2)^2} G^2.$$

*Proof.* Lemma 3 and Assumption 4 imply

$$\|\tilde{g}_t\|^2 = \|\mathcal{C}(g_t + e_t)\|^2$$

$$\leq \|\mathcal{C}(g_t + e_t) - (g_t + e_t) + (g_t + e_t)\|^2$$

$$\leq 2(q^2 + 1)\|g_t + e_t\|^2$$

$$\leq 4(q^2 + 1)\left(G^2 + \frac{4q_\omega^2}{(1 - q_\omega^2)^2} G^2\right)$$

$$= \frac{4(1 + q^2)(1 + q_\omega^2)^2}{(1 - q_\omega^2)^2} G^2.$$

It's then easy to show by the updating rule of $\hat{v}_t$, there exists a $j \in [t]$ such that $\hat{v}_{t,i} = v_{j,i}$. Then

$$\hat{v}_{t,i} = (1 - \beta_2) \sum_{\tau=1}^{j} \beta_2^{j-\tau} \tilde{g}_{\tau,i}^2 \leq \frac{4(1+q^2)(1+q_\omega^2)^2}{(1-q_\omega^2)^2} G^2,$$

which concludes the claim. $\qquad\square$

**Lemma 7.** *For* $D_t := \frac{1}{\sqrt{\hat{v}_{t-1}+\epsilon}} - \frac{1}{\sqrt{\hat{v}_t+\epsilon}}$ *we have*

$$\sum_{t=1}^{T} \|D_t\|_1 \leq \frac{d}{\sqrt{\epsilon}}, \quad \sum_{t=1}^{T} \|D_t\|^2 \leq \frac{d}{\epsilon}.$$

*Proof.* By the update rule, we have $\hat{v}_{t-1,i} \leq \hat{v}_{t,i}$ for any iterate $t$ and coordinate $i \in [d]$. Therefore, by the initialization $\hat{v}_0 = 0$, we get

$$\sum_{t=1}^{T} \|D_t\|_1 = \sum_{t=1}^{T} \sum_{i=1}^{d} \left( \frac{1}{\sqrt{\hat{v}_{t-1,i}+\epsilon}} - \frac{1}{\sqrt{\hat{v}_{t,i}+\epsilon}} \right) = \sum_{i=1}^{d} \left( \frac{1}{\sqrt{\hat{v}_{0,i}+\epsilon}} - \frac{1}{\sqrt{\hat{v}_{T,i}+\epsilon}} \right) \leq \frac{d}{\sqrt{\epsilon}}.$$

For the sum of squared $l_2$ norms, note the fact that for $a \geq b > 0$, it holds that

$$(a-b)^2 \leq (a-b)(a+b) = a^2 - b^2.$$

Thus,

$$\sum_{t=1}^{T} \|D_t\|^2 = \sum_{t=1}^{T} \sum_{i=1}^{d} \left( \frac{1}{\sqrt{\hat{v}_{t-1,i}+\epsilon}} - \frac{1}{\sqrt{\hat{v}_{t,i}+\epsilon}} \right)^2 \leq \sum_{t=1}^{T} \sum_{i=1}^{d} \left( \frac{1}{\hat{v}_{t-1,i}+\epsilon} - \frac{1}{\hat{v}_{t,i}+\epsilon} \right) \leq \frac{d}{\epsilon},$$

which gives the desired result. $\qquad\square$

## E.2 Non-convex Analysis

Here we derive the convergence rate with fixed step-size $\eta$. The rate shown in the main part can be obtained by plugging the expression of $\eta$ shown after the proof.

**Theorem 3. (Non-convex convergence rate)** *Let Assumptions 1, 2, 3, 4, 5 hold and* $q_\omega := (1+\omega)q < 1$. *Then, choosing any step-size* $\eta \leq \frac{\epsilon}{4LC_0}$, MICROADAM *(Algorithm 3) satisfies*

$$\frac{1}{T} \sum_{t=1}^{T} \mathbb{E}[\|\nabla f(\theta_t)\|^2] \leq 2C_0 \left( \frac{f(\theta_1) - f^*}{T\eta} + \frac{\eta L \sigma^2}{\epsilon} + \frac{\eta L C_2^2 G^2}{\epsilon} \right.$$
$$\left. + \frac{\eta^2 L^2 C_0 C_1^2 G^2}{\epsilon^2} + \frac{(1+C_1)G^2 d}{T\sqrt{\epsilon}} + \frac{\eta(1+2C_1)C_1 L G^2 d}{T\epsilon} \right),$$

*with constants* $C_0 := \sqrt{\frac{4(1+q_\omega^2)^3}{(1-q_\omega^2)^2} G^2 + \epsilon}$, $C_1 := \frac{\beta_1}{1-\beta_1}(1 + C_2) + \frac{2q_\omega}{1-q_\omega^2}$, $C_2 := \omega q (1 + \frac{2q_\omega}{1-q_\omega^2})$.

*Proof.* Similar to the proof of Comp-AMS [Li et al., 2022], we define two virtual iterates $\theta_t'$ and $x_t$.

$$\theta_{t+1}' := \theta_{t+1} - \eta \frac{\mathcal{E}_{t+1}}{\sqrt{\hat{v}_t + \epsilon}}$$
$$x_{t+1} := \theta_{t+1}' - \eta \frac{\beta_1}{1-\beta_1} \frac{m_t' + \mathcal{Z}_t}{\sqrt{\hat{v}_t + \epsilon}}.$$

Then, we derive the recurrence relation for each sequence as follows:

$$
\begin{aligned}
\theta'_{t+1} &= \theta_{t+1} - \eta \frac{\mathcal{E}_{t+1}}{\sqrt{\hat{v}_t} + \epsilon} \\
&= \theta_t - \eta \frac{(1 - \beta_1) \sum_{\tau=1}^{t} \beta_1^{t-\tau} \tilde{g}_\tau + (1 - \beta_1) \sum_{\tau=1}^{t+1} \beta_1^{t+1-\tau} e_\tau}{\sqrt{\hat{v}_t} + \epsilon} \\
&= \theta_t - \eta \frac{(1 - \beta_1) \sum_{\tau=1}^{t} \beta_1^{t-\tau} (\tilde{g}_\tau + e_{\tau+1}) + (1 - \beta) \beta_1^t e_1}{\sqrt{\hat{v}_t} + \epsilon} \\
&= \theta_t - \eta \frac{(1 - \beta_1) \sum_{\tau=1}^{t} \beta_1^{t-\tau} (g_\tau + e_\tau + \zeta_\tau)}{\sqrt{\hat{v}_t} + \epsilon} \\
&= \theta_t - \eta \frac{(1 - \beta_1) \sum_{\tau=1}^{t} \beta_1^{t-\tau} e_\tau}{\sqrt{\hat{v}_t} + \epsilon} - \eta \frac{m'_t}{\sqrt{\hat{v}_t} + \epsilon} - \eta \frac{\mathcal{Z}_t}{\sqrt{\hat{v}_t} + \epsilon} \\
&= \theta_t - \eta \frac{\mathcal{E}_t}{\sqrt{\hat{v}_{t-1}} + \epsilon} - \eta \frac{m'_t}{\sqrt{\hat{v}_t} + \epsilon} + \eta \left( \frac{1}{\sqrt{\hat{v}_{t-1}} + \epsilon} - \frac{1}{\sqrt{\hat{v}_t} + \epsilon} \right) \mathcal{E}_t - \eta \frac{\mathcal{Z}_t}{\sqrt{\hat{v}_t} + \epsilon} \\
&= \theta'_t - \eta \frac{m'_t}{\sqrt{\hat{v}_t} + \epsilon} + \eta \left( \frac{1}{\sqrt{\hat{v}_{t-1}} + \epsilon} - \frac{1}{\sqrt{\hat{v}_t} + \epsilon} \right) \mathcal{E}_t - \eta \frac{\mathcal{Z}_t}{\sqrt{\hat{v}_t} + \epsilon} \\
&= \theta'_t - \eta \frac{m'_t + \mathcal{Z}_t}{\sqrt{\hat{v}_t} + \epsilon} + \eta D_t \mathcal{E}_t,
\end{aligned}
$$

where we used the fact that $\tilde{g}_t + e_{t+1} = g_t + e_t + \zeta_t$ with quantization noise $\zeta_t$, and $e_0 = 0$ at initialization. Next, for the $x_t$ iterates we have

$$
\begin{aligned}
x_{t+1} &= \theta'_{t+1} - \eta \frac{\beta_1}{1 - \beta_1} \frac{m'_t + \mathcal{Z}_t}{\sqrt{\hat{v}_t} + \epsilon} \\
&= \theta'_t - \eta \frac{m'_t + \mathcal{Z}_t}{\sqrt{\hat{v}_t} + \epsilon} - \eta \frac{\beta_1}{1 - \beta_1} \frac{m'_t + \mathcal{Z}_t}{\sqrt{\hat{v}_t} + \epsilon} + \eta D_t \mathcal{E}_t \\
&= \theta'_t - \eta \frac{\beta_1 (m'_{t-1} + \mathcal{Z}_{t-1}) + (1 - \beta_1)(g_t + \zeta_t) + \frac{\beta_1^2}{1 - \beta_1} (m'_{t-1} + \mathcal{Z}_{t-1}) + \beta_1 (g_t + \zeta_t)}{\sqrt{\hat{v}_t} + \epsilon} \\
&\quad + \eta D_t \mathcal{E}_t \\
&= \theta'_t - \eta \frac{\beta_1}{1 - \beta_1} \frac{m'_{t-1} + \mathcal{Z}_{t-1}}{\sqrt{\hat{v}_t} + \epsilon} - \eta \frac{g_t + \zeta_t}{\sqrt{\hat{v}_t} + \epsilon} + \eta D_t \mathcal{E}_t \\
&= x_t - \eta \frac{g_t + \zeta_t}{\sqrt{\hat{v}_t} + \epsilon} + \eta \frac{\beta_1}{1 - \beta_1} D_t (m'_{t-1} + \mathcal{Z}_{t-1}) + \eta D_t \mathcal{E}_t.
\end{aligned}
$$

Next we apply smoothness (Assumption 3) of the loss function $f$ over the iterates $x_t$. From the gradient Lipschitzness we have

$$
f(x_{t+1}) \le f(x_t) + \langle \nabla f(x_t), x_{t+1} - x_t \rangle + \frac{L}{2} \| x_{t+1} - x_t \|^2.
$$

Due to unbiasedness of the compressor $\mathcal{Q}$ (see Assumption 2), we have $\mathbb{E}[\zeta_t | g_t, \tilde{g}_t, e_t, \hat{v}_t] = 0$. Taking expectation, we obtain

$$
\begin{aligned}
\mathbb{E}[f(x_{t+1})] - \mathbb{E}[f(x_t)] \quad \leq \quad & -\eta \mathbb{E}\left[\left\langle \nabla f(x_t), \frac{g_t + \zeta_t}{\sqrt{\hat{v}_t} + \epsilon} \right\rangle\right] \\
& + \eta \mathbb{E}\left[\left\langle \nabla f(x_t), \frac{\beta_1}{1 - \beta_1} D_t(m'_{t-1} + \mathcal{Z}_{t-1}) + D_t \mathcal{E}_t \right\rangle\right] \\
& + \frac{\eta^2 L}{2} \mathbb{E}\left[\left\| \frac{g_t + \zeta_t}{\sqrt{\hat{v}_t} + \epsilon} - \frac{\beta_1}{1 - \beta_1} D_t(m'_{t-1} + \mathcal{Z}_{t-1}) - D_t \mathcal{E}_t \right\|^2\right] \\
= \quad & \underbrace{-\eta \mathbb{E}\left[\left\langle \nabla f(\theta_t), \frac{g_t}{\sqrt{\hat{v}_t} + \epsilon} \right\rangle\right]}_{I} \qquad\qquad (3) \\
& + \underbrace{\eta \mathbb{E}\left[\left\langle \nabla f(x_t), \frac{\beta_1}{1 - \beta_1} D_t(m'_{t-1} + \mathcal{Z}_{t-1}) + D_t \mathcal{E}_t \right\rangle\right]}_{II} \\
& + \underbrace{\frac{\eta^2 L}{2} \mathbb{E}\left[\left\| \frac{g_t + \zeta_t}{\sqrt{\hat{v}_t} + \epsilon} - \frac{\beta_1}{1 - \beta_1} D_t(m'_{t-1} + \mathcal{Z}_{t-1}) - D_t \mathcal{E}_t \right\|^2\right]}_{III} \\
& + \underbrace{\eta \mathbb{E}\left[\left\langle \nabla f(\theta_t) - \nabla f(x_t), \frac{g_t}{\sqrt{\hat{v}_t} + \epsilon} \right\rangle\right]}_{IV}, \qquad\qquad (4)
\end{aligned}
$$

In the following, we bound all the four terms highlighted above.

**Bounding term I.** We have

$$
\begin{aligned}
I \quad = \quad & -\eta \mathbb{E}\left[\left\langle \nabla f(\theta_t), \frac{g_t}{\sqrt{\hat{v}_{t-1}} + \epsilon} \right\rangle\right] - \eta \mathbb{E}\left[\left\langle \nabla f(\theta_t), \left( \frac{1}{\sqrt{\hat{v}_t} + \epsilon} - \frac{1}{\sqrt{\hat{v}_{t-1}} + \epsilon} \right) g_t \right\rangle\right] \\
\leq \quad & -\eta \mathbb{E}\left[\left\langle \nabla f(\theta_t), \frac{\nabla f(\theta_t)}{\sqrt{\hat{v}_{t-1}} + \epsilon} \right\rangle\right] + \eta G^2 \mathbb{E}[\|D_t\|]. \\
\leq \quad & -\frac{\eta}{\sqrt{\frac{4(1+q_\omega^2)^3}{(1-q_\omega^2)^2} G^2} + \epsilon} \mathbb{E}[\|\nabla f(\theta_t)\|^2] + \eta G^2 \mathbb{E}[\|D_t\|_1], \qquad\qquad (5)
\end{aligned}
$$

where we use Assumption 4, Lemma 6 and the fact that $l_2$ norm is no larger than $l_1$ norm.

**Bounding term II.** By the definition of $\mathcal{E}_t$ and $\mathcal{Z}_t$, we know that

$$
\begin{aligned}
\|\mathcal{E}_t\| \quad \leq \quad & (1 - \beta_1) \sum_{\tau=1}^{t} \beta_1^{t-\tau} \|e_t\| \leq \frac{2q_\omega}{1 - q_\omega^2} G, \\
\|\mathcal{Z}_t\| \quad \leq \quad & (1 - \beta_1) \sum_{\tau=1}^{t} \beta_1^{t-\tau} \|\zeta_t\| \leq \omega q \left(1 + \frac{2q_\omega}{1 - q_\omega^2}\right) G.
\end{aligned}
$$

Then we have

$$II \leq \eta \mathbb{E}\left[\left\langle \nabla f(\theta_t), \frac{\beta_1}{1-\beta_1} D_t(m'_{t-1} + \mathcal{Z}_{t-1}) + D_t \mathcal{E}_t \right\rangle\right]$$

$$+ \eta \mathbb{E}\left[\left\langle \nabla f(x_t) - \nabla f(\theta_t), \frac{\beta_1}{1-\beta_1} D_t(m'_{t-1} + \mathcal{Z}_{t-1}) + D_t \mathcal{E}_t \right\rangle\right]$$

$$\leq \eta \mathbb{E}\left[\|\nabla f(\theta_t)\| \left\| \frac{\beta_1}{1-\beta_1} D_t(m'_{t-1} + \mathcal{Z}_{t-1}) + D_t \mathcal{E}_t \right\|\right]$$

$$+ \eta^2 L \mathbb{E}\left[\left\| \frac{\frac{\beta_1}{1-\beta_1} m'_{t-1} + \frac{\beta_1}{1-\beta_1}\mathcal{Z}_{t-1} + \mathcal{E}_t}{\sqrt{\hat{v}_{t-1} + \epsilon}} \right\| \left\| \frac{\beta_1}{1-\beta_1} D_t(m'_{t-1} + \mathcal{Z}_{t-1}) + D_t \mathcal{E}_t \right\|\right]$$

$$\leq \eta C_1 G^2 \mathbb{E}[\|D_t\|_1] + \frac{\eta^2 C_1^2 L G^2}{\sqrt{\epsilon}}\mathbb{E}[\|D_t\|_1], \tag{6}$$

where $C_1 = \frac{\beta_1}{1-\beta_1}\left(1 + \omega q\left(1 + \frac{2q_\omega}{1-q_\omega^2}\right)\right) + \frac{2q_\omega}{1-q_\omega^2}$. The second inequality is because of smoothness of $f(\theta)$, and the last inequality is due to Lemma 3, Assumption 4 and the property of norms.

**Bounding term III.** This term can be bounded as follows:

$$III \leq \eta^2 L \mathbb{E}\left[\left\| \frac{g_t + \zeta_t}{\sqrt{\hat{v}_t + \epsilon}} \right\|^2\right] + \eta^2 L \mathbb{E}\left[\left\| \frac{\beta_1}{1-\beta_1} D_t(m'_{t-1} + \mathcal{Z}_{t-1}) - D_t \mathcal{E}_t \right\|^2\right]$$

$$\leq \frac{2\eta^2 L}{\epsilon}\mathbb{E}[\|g_t - \nabla f(\theta_t) + \nabla f(\theta_t)\|^2] + \frac{2\eta^2 L}{\epsilon}\mathbb{E}[\|\zeta_t\|^2] \tag{7}$$

$$+ \eta^2 L \mathbb{E}\left[\left\| D_t\left(\frac{\beta_1}{1-\beta_1} m'_{t-1} + \frac{\beta_1}{1-\beta_1}\mathcal{Z}_{t-1} - \mathcal{E}_t\right)\right\|^2\right]$$

$$\leq \frac{2\eta^2 L}{\epsilon}\mathbb{E}[\|\nabla f(\theta_t)\|^2] + \frac{2\eta^2 L\sigma^2}{\epsilon} + \frac{2\eta^2 L}{\epsilon}\omega^2 q^2 \left(1 + \frac{2q}{1-q^2}\right)^2 G^2 + \eta^2 C_1^2 L G^2 \mathbb{E}[\|D_t\|^2]$$

$$\leq \frac{2\eta^2 L}{\epsilon}\mathbb{E}[\|\nabla f(\theta_t)\|^2] + \frac{2\eta^2 L(\sigma^2 + C_2^2 G^2)}{\epsilon} + \eta^2 C_1^2 L G^2 \mathbb{E}[\|D_t\|^2], \tag{8}$$

where $C_2 = \omega q(1 + \frac{2q}{1-q^2})$ and we used Assumption 5 that $g_t$ is unbiased with bounded variance $\sigma^2$.

**Bounding term IV.** We have

$$IV = \eta \mathbb{E}\left[\left\langle \nabla f(\theta_t) - \nabla f(x_t), \frac{g_t}{\sqrt{\hat{v}_{t-1} + \epsilon}} \right\rangle\right] \tag{9}$$

$$+ \eta \mathbb{E}\left[\left\langle \nabla f(\theta_t) - \nabla f(x_t), \left(\frac{1}{\sqrt{\hat{v}_t + \epsilon}} - \frac{1}{\sqrt{\hat{v}_{t-1} + \epsilon}}\right) g_t \right\rangle\right]$$

$$\leq \eta \mathbb{E}\left[\left\langle \nabla f(\theta_t) - \nabla f(x_t), \frac{\nabla f(\theta_t)}{\sqrt{\hat{v}_{t-1} + \epsilon}} \right\rangle\right] \tag{10}$$

$$+ \eta^2 L \mathbb{E}\left[\left\| \frac{\frac{\beta_1}{1-\beta_1} m'_{t-1} + \frac{\beta_1}{1-\beta_1}\mathcal{Z}_{t-1} + \mathcal{E}_t}{\sqrt{\hat{v}_{t-1} + \epsilon}} \right\| \|D_t g_t\|\right]$$

$$\overset{(a)}{\leq} \frac{\eta\rho}{2\epsilon}\mathbb{E}[\|\nabla f(\theta_t)\|^2] + \frac{\eta}{2\rho}\mathbb{E}[\|\nabla f(\theta_t) - \nabla f(x_t)\|^2] + \frac{\eta^2 C_1 L G^2}{\sqrt{\epsilon}}\mathbb{E}[\|D_t\|]$$

$$\overset{(b)}{\leq} \frac{\eta\rho}{2\epsilon}\mathbb{E}[\|\nabla f(\theta_t)\|^2] + \frac{\eta^3 L^2}{2\rho}\mathbb{E}\left[\left\| \frac{\frac{\beta_1}{1-\beta_1} m'_{t-1} + \frac{\beta_1}{1-\beta_1}\mathcal{Z}_{t-1} + \mathcal{E}_t}{\sqrt{\hat{v}_{t-1} + \epsilon}} \right\|^2\right] + \frac{\eta^2 C_1 L G^2}{\sqrt{\epsilon}}\mathbb{E}[\|D_t\|_1]$$

$$\leq \frac{\eta\rho}{2\epsilon}\mathbb{E}[\|\nabla f(\theta_t)\|^2] + \frac{\eta^3 L^2}{2\rho}\frac{C_1^2 G^2}{\epsilon} + \frac{\eta^2 L C_1 G^2}{\sqrt{\epsilon}}\mathbb{E}[\|D_t\|_1], \tag{11}$$

where (a) is due to Young's inequality and (b) is based on Assumption 3. Now integrating (5), (6), (8), (11) into (4),

$$I \leq -\frac{\eta}{C_0}\mathbb{E}[\|\nabla f(\theta_t)\|^2] + \eta G^2\mathbb{E}[\|D_t\|_1]$$

$$II \leq \eta C_1 G^2\mathbb{E}[\|D_t\|_1] + \frac{\eta^2 C_1^2 L G^2}{\sqrt{\epsilon}}\mathbb{E}[\|D_t\|_1]$$

$$III \leq \frac{\eta^2 L}{\epsilon}\mathbb{E}[\|\nabla f(\theta_t)\|^2] + \frac{\eta^2 L(\sigma^2 + C_2^2 G^2)}{\epsilon} + \eta^2 C_1^2 L G^2\mathbb{E}[\|D_t\|^2]$$

$$IV \leq \frac{\eta\rho}{2\epsilon}\mathbb{E}[\|\nabla f(\theta_t)\|^2] + \frac{\eta^3 L^2}{2\rho}\frac{C_1^2 G^2}{\epsilon} + \frac{\eta^2 L C_1 G^2}{\sqrt{\epsilon}}\mathbb{E}[\|D_t\|_1],$$

and taking the telescoping summation over $t = 1, \ldots, T$, we obtain

$$\mathbb{E}[f(x_{T+1}) - f(x_1)]$$
$$\leq \left(-\frac{\eta}{C_0} + \frac{\eta^2 L}{\epsilon} + \frac{\eta\rho}{2\epsilon}\right)\sum_{t=1}^{T}\mathbb{E}[\|\nabla f(\theta_t)\|^2] + \frac{T\eta^2 L(\sigma^2 + C_2^2 G^2)}{\epsilon} + \frac{T\eta^3 L^2 C_1^2 G^2}{2\rho\epsilon}$$
$$+ \left(\eta(1 + C_1)G^2 + \frac{\eta^2(1 + C_1)C_1 L G^2}{\sqrt{\epsilon}}\right)\sum_{t=1}^{T}\mathbb{E}[\|D_t\|_1] + \eta^2 C_1^2 L G^2\sum_{t=1}^{T}\mathbb{E}[\|D_t\|^2].$$

Setting $\eta \leq \frac{\epsilon}{4LC_0}$ and choosing $\rho = \frac{\epsilon}{2C_0}$, we further arrive at

$$\mathbb{E}[f(x_{T+1}) - f(x_1)] \leq -\frac{\eta}{2C_0}\sum_{t=1}^{T}\mathbb{E}[\|\nabla f(\theta_t)\|^2] + \frac{T\eta^2 L(\sigma^2 + C_2^2 G^2)}{\epsilon}$$
$$+ \frac{T\eta^3 L^2 C_0 C_1^2 G^2}{\epsilon^2} + \frac{\eta(1 + C_1)G^2 d}{\sqrt{\epsilon}} + \frac{\eta^2(1 + 2C_1)C_1 L G^2 d}{\epsilon}.$$

where the inequality follows from Lemma 7. Re-arranging terms, we get that

$$\frac{1}{T}\sum_{t=1}^{T}\mathbb{E}[\|\nabla f(\theta_t)\|^2]$$
$$\leq 2C_0\left(\frac{\mathbb{E}[f(x_1) - f(x_{T+1})]}{T\eta} + \frac{\eta L(\sigma^2 + C_2^2 G^2)}{\epsilon} + \frac{\eta^2 L^2 C_0 C_1^2 G^2}{\epsilon^2}\right)$$
$$+ 2C_0\left(\frac{(1 + C_1)G^2 d}{T\sqrt{\epsilon}} + \frac{\eta(1 + 2C_1)C_1 L G^2 d}{T\epsilon}\right)$$
$$\leq 2C_0\left(\frac{f(\theta_1) - f^*}{T\eta} + \frac{\eta L(\sigma^2 + C_2^2 G^2)}{\epsilon} + \frac{\eta^2 L^2 C_0 C_1^2 G^2}{\epsilon^2}\right)$$
$$+ 2C_0\left(\frac{(1 + C_1)G^2 d}{T\sqrt{\epsilon}} + \frac{\eta(1 + 2C_1)C_1 L G^2 d}{T\epsilon}\right),$$

where in the last inequality we used $x_1 = \theta_1$ and the lower bound $f^* \leq f(\theta)$ for all $\theta \in \mathbb{R}^d$. $\qquad\square$

To get the rate mentioned in the main part, choose $\eta = \min\{\frac{\epsilon}{4LC_0}, \frac{1}{\sqrt{T}}\}$ and continue

$$\frac{1}{T}\sum_{t=1}^{T}\mathbb{E}[\|\nabla f(\theta_t)\|^2]$$

$$\leq 2C_0\left(\max\left\{1, \frac{4LC_0}{\epsilon\sqrt{T}}\right\}\frac{f(\theta_1)-f^*}{\sqrt{T}} + \frac{L(\sigma^2+C_2^2G^2)}{\epsilon\sqrt{T}} + \frac{L^2C_0C_1^2G^2}{\epsilon^2 T}\right)$$

$$+ 2C_0\left(\frac{(1+C_1)G^2d}{T\sqrt{\epsilon}} + \frac{(1+2C_1)C_1LG^2d}{\epsilon T^{3/2}}\right)$$

$$\leq 2C_0\left(\frac{f(\theta_1)-f^*}{\sqrt{T}} + \frac{L(\sigma^2+C_2^2G^2)}{\epsilon\sqrt{T}}\right)$$

$$+ 2C_0\left(\frac{4LC_0}{\epsilon}\frac{f(\theta_1)-f^*}{T} + \frac{L^2C_0C_1^2G^2}{\epsilon^2 T} + \frac{(1+C_1)G^2d}{T\sqrt{\epsilon}} + \frac{(1+2C_1)C_1LG^2d}{\epsilon T^{3/2}}\right)$$

$$= 2C_0\left(\frac{f(\theta_1)-f^*}{\sqrt{T}} + \frac{L(\sigma^2+C_2^2G^2)}{\epsilon\sqrt{T}}\right) + \mathcal{O}\left(\frac{G^3(G+d)}{T}\right),$$

where in the second part of the rate we suppressed all the problem and compression dependent constants.

### E.3    Analysis Under PL Condition

As in the non-convex analysis, here we derive the convergence rate with fixed step-size $\eta$. The rate shown in the main part can be obtained by plugging the expression of $\eta$.

**Theorem 4. (Convergence rate under PL)** *Let Assumptions 1, 2, 3, 4, 5 and 6 hold, and $q_\omega < 1$. Then, choosing any step-size $\eta \leq \frac{\epsilon}{4LC_0}$, MICROADAM (Algorithm 3) satisfies*

$$\mathbb{E}[f(\theta_{T+1})] - f^* \leq \left(1 - \frac{\eta\mu}{C_0}\right)^T(f(\theta_1)-f^*) + \eta\left(\frac{LC_0\sigma^2+LC_0(C_1+C_2^2)G^2}{\mu\epsilon} + \frac{(1+C_1)G^2d+C_1G^2}{\sqrt{\epsilon}}\right)$$

$$+ \eta^2\left(\frac{3L^2C_0C_1^2G^2}{2\mu\epsilon^{3/2}} + \frac{(1+2C_1)C_1LG^2d}{\epsilon} + \frac{LC_1^2G^2}{2\epsilon}\right).$$

*Proof.* We start from descent lemma

$$\mathbb{E}[f(x_{t+1})] - f(x_t)$$

$$\leq -\eta\mathbb{E}\left[\left\langle\nabla f(x_t), \frac{g_t+\zeta_t}{\sqrt{\hat{v}_t}+\epsilon}\right\rangle\right] + \eta\mathbb{E}\left[\left\langle\nabla f(x_t), \frac{\beta_1}{1-\beta_1}D_t(m'_{t-1}+\mathcal{Z}_{t-1}) + D_t\mathcal{E}_t\right\rangle\right]$$

$$+ \frac{\eta^2 L}{2}\mathbb{E}\left[\left\|\frac{g_t+\zeta_t}{\sqrt{\hat{v}_t}+\epsilon} - \frac{\beta_1}{1-\beta_1}D_t(m'_{t-1}+\mathcal{Z}_{t-1}) - D_t\mathcal{E}_t\right\|^2\right]$$

$$= \underbrace{-\eta\mathbb{E}\left[\left\langle\nabla f(x_t), \frac{g_t}{\sqrt{\hat{v}_t}+\epsilon}\right\rangle\right]}_{I'} + \underbrace{\eta\mathbb{E}\left[\left\langle\nabla f(x_t), \frac{\beta_1}{1-\beta_1}D_t(m'_{t-1}+\mathcal{Z}_{t-1}) + D_t\mathcal{E}_t\right\rangle\right]}_{II}$$

$$+ \underbrace{\frac{\eta^2 L}{2}\mathbb{E}\left[\left\|\frac{g_t+\zeta_t}{\sqrt{\hat{v}_t}+\epsilon} - \frac{\beta_1}{1-\beta_1}D_t(m'_{t-1}+\mathcal{Z}_{t-1}) - D_t\mathcal{E}_t\right\|^2\right]}_{III}. \quad (12)$$

We bound part $II$ and part $III$ in the same way as it was done in the non-convex analysis. We now provide a bound for part $I'$:

$$
\begin{aligned}
I' = & -\eta\mathbb{E}\left[\left\langle \nabla f(x_t), \frac{g_t}{\sqrt{\hat{v}_t + \epsilon}}\right\rangle\right] \\
= & -\eta\mathbb{E}\left[\left\langle \nabla f(x_t), \frac{g_t}{\sqrt{\hat{v}_{t-1} + \epsilon}}\right\rangle\right] - \eta\mathbb{E}\left[\left\langle \nabla f(x_t), g_t\left(\frac{1}{\sqrt{\hat{v}_t + \epsilon}} - \frac{1}{\sqrt{\hat{v}_{t-1} + \epsilon}}\right)\right\rangle\right] \\
= & -\eta\mathbb{E}\left[\left\langle \nabla f(x_t), \frac{g_t - \nabla f(x_t) + \nabla f(x_t)}{\sqrt{\hat{v}_{t-1} + \epsilon}}\right\rangle\right] \\
& -\eta\mathbb{E}\left[\left\langle \nabla f(x_t), g_t\left(\frac{1}{\sqrt{\hat{v}_t + \epsilon}} - \frac{1}{\sqrt{\hat{v}_{t-1} + \epsilon}}\right)\right\rangle\right] \\
= & -\eta\mathbb{E}\left[\left\langle \nabla f(x_t), \frac{\nabla f(x_t)}{\sqrt{\hat{v}_{t-1} + \epsilon}}\right\rangle\right] - \eta\mathbb{E}\left[\left\langle \nabla f(x_t), \frac{g_t - \nabla f(x_t)}{\sqrt{\hat{v}_{t-1} + \epsilon}}\right\rangle\right] \\
& -\eta\mathbb{E}\left[\left\langle \nabla f(x_t), g_t\left(\frac{1}{\sqrt{\hat{v}_t + \epsilon}} - \frac{1}{\sqrt{\hat{v}_{t-1} + \epsilon}}\right)\right\rangle\right].
\end{aligned}
$$

We further expand and bound this equation as follows:

$$
\begin{aligned}
I' \leq & -\frac{\eta}{C_0}\mathbb{E}\left[\|\nabla f(x_t)\|^2\right] \\
& -\eta\mathbb{E}\left[\left\langle \nabla f(x_t) - \nabla f(\theta_t) + \nabla f(\theta_t), \frac{1}{\sqrt{\hat{v}_{t-1} + \epsilon}}\left(\nabla f(\theta_t) - \nabla f(x_t)\right)\right\rangle\right] \\
& -\eta\mathbb{E}\left[\left\langle \nabla f(x_t) - \nabla f(\theta_t) + \nabla f(\theta_t), \left(\frac{1}{\sqrt{\hat{v}_t + \epsilon}} - \frac{1}{\sqrt{\hat{v}_{t-1} + \epsilon}}\right)g_t\right\rangle\right] \\
= & -\frac{\eta}{C_0}\mathbb{E}\left[\|\nabla f(x_t)\|^2\right] \\
& -\eta\mathbb{E}\left[\left\langle \nabla f(x_t) - \nabla f(\theta_t), \frac{1}{\sqrt{\hat{v}_{t-1} + \epsilon}}\left(\nabla f(\theta_t) - \nabla f(x_t)\right)\right\rangle\right] \\
& -\eta\mathbb{E}\left[\left\langle \nabla f(\theta_t), \frac{1}{\sqrt{\hat{v}_{t-1} + \epsilon}}\left(\nabla f(\theta_t) - \nabla f(x_t)\right)\right\rangle\right] \\
& -\eta\mathbb{E}\left[\left\langle \nabla f(x_t) - \nabla f(\theta_t), \left(\frac{1}{\sqrt{\hat{v}_t + \epsilon}} - \frac{1}{\sqrt{\hat{v}_{t-1} + \epsilon}}\right)g_t\right\rangle\right] \\
& -\eta\mathbb{E}\left[\left\langle \nabla f(\theta_t), \left(\frac{1}{\sqrt{\hat{v}_t + \epsilon}} - \frac{1}{\sqrt{\hat{v}_{t-1} + \epsilon}}\right)g_t\right\rangle\right] \\
\leq & -\frac{\eta}{C_0}\mathbb{E}\left[\|\nabla f(x_t)\|^2\right] + \frac{\eta}{\sqrt{\epsilon}}\mathbb{E}\left[\|\nabla f(x_t) - f(\theta_t)\|^2\right] \\
& -\eta\mathbb{E}\left[\left\langle \nabla f(\theta_t), \frac{1}{\sqrt{\hat{v}_{t-1} + \epsilon}}\left(\nabla f(\theta_t) - \nabla f(x_t)\right)\right\rangle\right] \\
& +\eta\mathbb{E}\left[\langle \nabla f(x_t) - \nabla f(\theta_t), D_t g_t\rangle\right] + \eta\mathbb{E}\left[\langle \nabla f(\theta_t), D_t g_t\rangle\right].
\end{aligned}
$$

Next, we use the Cauchy–Schwartz inequality to bound inner products above, $L$-smoothness inequality to bound $\|\nabla f(x_t) - \nabla f(\theta_t)\| \leq L\|x_t - \theta_t\| \leq \frac{\eta L C_1 G}{\sqrt{\epsilon}}$, and the inequality $-\|a\|^2 \leq -\frac{1}{2}\|b\|^2 +$

$\|a - b\|^2$ for the first term:

$$
\begin{aligned}
I' &\leq -\frac{\eta}{C_0}\mathbb{E}\left[\|\nabla f(x_t)\|^2\right] + \frac{\eta}{\sqrt{\epsilon}}\mathbb{E}\left[\|\nabla f(x_t) - f(\theta_t)\|^2\right] + \frac{\eta G}{\sqrt{\epsilon}}\mathbb{E}\left[\|\nabla f(\theta_t) - \nabla f(x_t)\|\right] \\
&\quad + \eta G\mathbb{E}\left[\|\nabla f(x_t) - \nabla f(\theta_t)\|\|D_t\|\right] + \eta G^2\mathbb{E}\left[\|D_t\|\right] \\
&\leq -\frac{\eta}{2C_0}\mathbb{E}\left[\|\nabla f(x_t)\|^2\right] - \frac{\eta}{2C_0}\mathbb{E}\left[\|\nabla f(x_t)\|^2\right] + \frac{\eta}{\sqrt{\epsilon}}\frac{\eta^2 L^2 C_1^2 G^2}{\epsilon} + \frac{\eta^2 LC_1 G^2}{\epsilon} \\
&\quad + \eta G\frac{\eta LC_1 G}{\sqrt{\epsilon}}\mathbb{E}\left[\|D_t\|_1\right] + \eta G^2\mathbb{E}\left[\|D_t\|_1\right] \\
&\leq -\frac{\eta}{2C_0}\mathbb{E}\left[\|\nabla f(x_t)\|^2\right] - \frac{\eta}{4C_0}\mathbb{E}[\|\nabla f(\theta_t)\|^2] + \frac{\eta}{2C_0}\mathbb{E}[\|\nabla f(x_t) - \nabla f(\theta_t)\|^2] \\
&\quad + \frac{\eta^3 L^2 C_1^2 G^2}{\epsilon^{3/2}} + \frac{\eta^2 LC_1 G^2}{\epsilon} + \frac{\eta^2 LC_1 G^2}{\sqrt{\epsilon}}\mathbb{E}\left[\|D_t\|_1\right] + \eta G^2\mathbb{E}\left[\|D_t\|_1\right] \\
&\leq -\frac{\eta}{2C_0}\mathbb{E}\left[\|\nabla f(x_t)\|^2\right] - \frac{\eta}{4C_0}\mathbb{E}[\|\nabla f(\theta_t)\|^2] \\
&\quad + \frac{3\eta^3 L^2 C_1^2 G^2}{2\epsilon^{3/2}} + \frac{\eta^2 LC_1 G^2}{\epsilon} + \frac{\eta^2 LC_1 G^2}{\sqrt{\epsilon}}\mathbb{E}\left[\|D_t\|_1\right] + \eta G^2\mathbb{E}\left[\|D_t\|_1\right].
\end{aligned}
$$

Plugging the obtained bound for $I'$ with previously obtained bounds for $II$ and $III$

$$
\begin{aligned}
II &\leq \eta C_1 G^2\mathbb{E}[\|D_t\|_1] + \frac{\eta^2 C_1^2 LG^2}{\sqrt{\epsilon}}\mathbb{E}[\|D_t\|_1] \\
III &\leq \frac{\eta^2 L}{\epsilon}\mathbb{E}[\|\nabla f(\theta_t)\|^2] + \frac{\eta^2 L(\sigma^2 + C_2^2 G^2)}{\epsilon} + \eta^2 C_1^2 LG^2\mathbb{E}[\|D_t\|^2]
\end{aligned}
$$

into (12) and using the step-size bound $\eta \leq \frac{\epsilon}{4LC_0}$ we get

$$
\begin{aligned}
\mathbb{E}[f(x_{t+1})] - \mathbb{E}[f(x_t)] &\leq -\frac{\eta}{2C_0}\mathbb{E}\left[\|\nabla f(x_t)\|^2\right] - \frac{\eta}{4C_0}\mathbb{E}[\|\nabla f(\theta_t)\|^2] \\
&\quad + \frac{3\eta^3 L^2 C_1^2 G^2}{2\epsilon^{3/2}} + \frac{\eta^2 LC_1 G^2}{\epsilon} + \frac{\eta^2 LC_1 G^2}{\sqrt{\epsilon}}\mathbb{E}\left[\|D_t\|_1\right] \\
&\quad + \eta C_1 G^2\mathbb{E}[\|D_t\|_1] + \frac{\eta^2 C_1^2 LG^2}{\sqrt{\epsilon}}\mathbb{E}[\|D_t\|_1] + \eta G^2\mathbb{E}\left[\|D_t\|_1\right] \\
&\quad + \frac{\eta^2 L}{\epsilon}\mathbb{E}[\|\nabla f(\theta_t)\|^2] + \frac{\eta^2 L(\sigma^2 + C_2^2 G^2)}{\epsilon} + \eta^2 C_1^2 LG^2\mathbb{E}[\|D_t\|^2] \\
&\leq -\frac{\eta}{2C_0}\mathbb{E}\left[\|\nabla f(x_t)\|^2\right] + \frac{\eta^2 L\sigma^2}{\epsilon} + \frac{\eta^2 L(C_1 + C_2^2)G^2}{\epsilon} \\
&\quad + \eta(1 + C_1)G^2\mathbb{E}[\|D_t\|_1] + \frac{3\eta^3 L^2 C_1^2 G^2}{2\epsilon^{3/2}} \\
&\quad + \frac{\eta^2(1 + C_1)C_1 LG^2}{\sqrt{\epsilon}}\mathbb{E}[\|D_t\|_1] + \eta^2 C_1^2 LG^2\mathbb{E}[\|D_t\|^2] \\
&\leq -\frac{\eta\mu}{C_0}(\mathbb{E}[f(x_t)] - f^*) + \frac{\eta^2 L\sigma^2}{\epsilon} + \frac{\eta^2 L(C_1 + C_2^2)G^2}{\epsilon} \\
&\quad + \eta(1 + C_1)G^2\mathbb{E}[\|D_t\|_1] + \frac{\eta^2(1 + C_1)C_1 LG^2}{\sqrt{\epsilon}}\mathbb{E}[\|D_t\|_1] \\
&\quad + \eta^2 C_1^2 LG^2\mathbb{E}[\|D_t\|^2] + \frac{3\eta^3 L^2 C_1^2 G^2}{2\epsilon^{3/2}},
\end{aligned}
$$

where in the last inequality we applied PL condition from Assumption 6. After some reshuffling of the terms, we obtain the following recursion:

$$\mathbb{E}[f(x_{t+1})] - f^* \leq \left(1 - \frac{\eta\mu}{C_0}\right)(\mathbb{E}[f(x_t)] - f^*) + \frac{\eta^2 L\sigma^2}{\epsilon} + \frac{\eta^2 L(C_1 + C_2^2)G^2}{\epsilon} + \frac{3\eta^3 L^2 C_1^2 G^2}{2\epsilon^{3/2}}$$

$$+ \eta(1 + C_1)G^2 \mathbb{E}[\|D_t\|_1] + \frac{\eta^2(1 + C_1)C_1 LG^2}{\sqrt{\epsilon}}\mathbb{E}[\|D_t\|_1]$$

$$+ \eta^2 C_1^2 LG^2 \mathbb{E}[\|D_t\|^2].$$

Notice that $\eta \leq \frac{\epsilon}{4LC_0} \leq \frac{C_0}{4\mu}$, so that the coefficient $1 - \frac{\eta\mu}{C_0} \in (0, 1)$. Unrolling the recursion, we arrive

$$\mathbb{E}[f(x_{T+1})] - f^* \leq \left(1 - \frac{\eta\mu}{C_0}\right)^T (\mathbb{E}[f(x_1)] - f^*)$$

$$+ \left(\frac{\eta^2 L\sigma^2}{\epsilon} + \frac{\eta^2 L(C_1 + C_2^2)G^2}{\epsilon} + \frac{3\eta^3 L^2 C_1^2 G^2}{2\epsilon^{3/2}}\right)\sum_{t=1}^T \left(1 - \frac{\eta\mu}{C_0}\right)^t$$

$$+ \eta(1 + C_1)G^2 \sum_{t=1}^T \left(1 - \frac{\eta\mu}{C_0}\right)^t \mathbb{E}[\|D_t\|_1]$$

$$+ \frac{\eta^2(1 + C_1)C_1 LG^2}{\sqrt{\epsilon}} \sum_{t=1}^T \left(1 - \frac{\eta\mu}{C_0}\right)^t \mathbb{E}[\|D_t\|_1]$$

$$+ \eta^2 C_1^2 LG^2 \sum_{t=1}^T \left(1 - \frac{\eta\mu}{C_0}\right)^t \mathbb{E}[\|D_t\|^2]. \tag{13}$$

For the second sum above we upper bound it by its infinite sum as

$$\sum_{t=1}^T \left(1 - \frac{\eta\mu}{C_0}\right)^t \leq \sum_{t=0}^\infty \left(1 - \frac{\eta\mu}{C_0}\right)^t = \frac{C_0}{\eta\mu}.$$

For the other three sums we bound $1 - \frac{\eta\mu}{C_0} \leq 1$ and apply the bounds in Lemma 7:

$$\sum_{t=1}^T \left(1 - \frac{\eta\mu}{C_0}\right)^t \mathbb{E}[\|D_t\|_1] \leq \sum_{t=1}^T \mathbb{E}[\|D_t\|_1] \leq \frac{d}{\sqrt{\epsilon}},$$

$$\sum_{t=1}^T \left(1 - \frac{\eta\mu}{C_0}\right)^t \mathbb{E}[\|D_t\|^2] \leq \sum_{t=1}^T \mathbb{E}[\|D_t\|^2] \leq \frac{d}{\epsilon}.$$

Plugging all this bounds into (13) and noticing that $x_1 = \theta_1$, we finally get

$$\mathbb{E}[f(x_{T+1})] - f^* \leq \left(1 - \frac{\eta\mu}{C_0}\right)^T (f(\theta_1) - f^*)$$

$$+ \frac{C_0}{\eta\mu}\left(\frac{\eta^2 L\sigma^2}{\epsilon} + \frac{\eta^2 L(C_1 + C_2^2)G^2}{\epsilon} + \frac{3\eta^3 L^2 C_1^2 G^2}{2\epsilon^{3/2}}\right)$$

$$+ \frac{\eta(1 + C_1)G^2 d}{\sqrt{\epsilon}} + \frac{\eta^2(1 + C_1)C_1 LG^2 d}{\epsilon} + \frac{\eta^2 C_1^2 LG^2 d}{\epsilon}$$

$$\leq \left(1 - \frac{\eta\mu}{C_0}\right)^T (f(\theta_1) - f^*)$$

$$+ \eta\left(\frac{LC_0\sigma^2}{\mu\epsilon} + \frac{LC_0(C_1 + C_2^2)G^2}{\mu\epsilon} + \frac{(1 + C_1)G^2 d}{\sqrt{\epsilon}}\right)$$

$$+ \eta^2\left(\frac{3L^2 C_0 C_1^2 G^2}{2\mu\epsilon^{3/2}} + \frac{(1 + C_1)C_1 LG^2 d}{\epsilon} + \frac{C_1^2 LG^2 d}{\epsilon}\right).$$

The obtained rate above is with respect to the virtual iterates $x_t$ that we defined for the purposes of analysis. To convert this rate with respect to the iterates $\theta_t$ of the algorithm, we apply $L$-smoothness to bound the functional difference:

$$|f(x_t) - f(\theta_t)| \leq |\langle \nabla f(\theta_t), x_t - \theta_t \rangle| + \frac{L}{2} \|x_t - \theta_t\|^2 \leq \frac{\eta C_1 G^2}{\sqrt{\epsilon}} + \frac{\eta^2 L C_1^2 G^2}{2\epsilon},$$

which implies

$$\begin{aligned}
\mathbb{E}[f(\theta_{T+1})] - f^* \leq &\left(1 - \frac{\eta\mu}{C_0}\right)^T (f(\theta_1) - f^*) \\
&+ \eta \left( \frac{LC_0\sigma^2}{\mu\epsilon} + \frac{LC_0(C_1 + C_2^2)G^2}{\mu\epsilon} + \frac{(1+C_1)G^2 d}{\sqrt{\epsilon}} + \frac{C_1 G^2}{\sqrt{\epsilon}} \right) \\
&+ \eta^2 \left( \frac{3L^2 C_0 C_1^2 G^2}{2\mu\epsilon^{3/2}} + \frac{(1+C_1)C_1 LG^2 d}{\epsilon} + \frac{C_1^2 LG^2 d}{\epsilon} + \frac{LC_1^2 G^2}{2\epsilon} \right)
\end{aligned}$$

and completes the proof. $\qquad\qquad\qquad\qquad\qquad\qquad\qquad\qquad\qquad\qquad\qquad\qquad\qquad\quad\square$

To get the rate mentioned in the main part of the paper, we plug in the expression $\eta = \min\{\frac{\epsilon}{4LC_0}, \frac{2C_0 \log T}{\mu T}\}$ and collect higher order terms.

$$\begin{aligned}
\mathbb{E}[f(\theta_{T+1})] - f^* \leq &\max\left\{ \frac{1}{T^2}, \left(1 - \frac{\epsilon\mu}{4L}\right)^T \right\} (f(\theta_1) - f^*) \\
&+ \frac{\log T}{T} \frac{2C_0}{\mu} \left( \frac{LC_0\sigma^2}{\mu\epsilon} + \frac{LC_0(C_1 + C_2^2)G^2}{\mu\epsilon} + \frac{(1+C_1)G^2 d}{\sqrt{\epsilon}} + \frac{C_1 G^2}{\sqrt{\epsilon}} \right) \\
&+ \frac{\log^2 T}{T^2} \frac{4C_0^2}{\mu^2} \left( \frac{3L^2 C_0 C_1^2 G^2}{2\mu\epsilon^{3/2}} + \frac{(1+C_1)C_1 LG^2 d}{\epsilon} + \frac{C_1^2 LG^2 d}{\epsilon} + \frac{LC_1^2 G^2}{2\epsilon} \right) \\
\leq &\frac{2\log T}{T} \left( \frac{LC_0^2}{\mu} \frac{\sigma^2 + (C_1 + C_2^2)G^2}{\mu\epsilon} + \frac{C_0(1+C_1)(1+d)G^2}{\mu\sqrt{\epsilon}} \right) + \widetilde{\mathcal{O}}\left( \frac{G^4(G+d)}{T^2} \right).
\end{aligned}$$

### E.4    Non-convex Analysis with Weight Decay

---

**Algorithm 4** MICROADAMW (MICROADAM with Weight Decay)

---

1: Input: parameters $\beta_1$, $\beta_2 \in (0,1)$, $\epsilon > 0$, step-size $\eta > 0$, $\theta_1 \in \mathbb{R}^d$, $e_1 = m_0 = v_0 = \hat{v}_0 = 0_d$
2: **for** $t = \{1, 2, ..., T\}$ **do**
3:     $\quad g_t = \widetilde{\nabla}_\theta f(\theta_t)$ $\qquad\qquad\qquad\qquad\qquad\qquad\qquad\qquad$ $\diamond$ Compute unbiased stochastic gradient
4:     $\quad \tilde{g}_t = \mathcal{C}(g_t + e_t)$ $\qquad\qquad\qquad\qquad\qquad\qquad\qquad$ $\diamond$ Add accumulated error $e_t$ and compress
5:     $\quad e_{t+1} = \mathcal{Q}(e_t + g_t - \tilde{g}_t)$ $\qquad\qquad\qquad\qquad\qquad$ $\diamond$ Update and compress the error
6:     $\quad m_t = \beta_1 m_{t-1} + (1 - \beta_1)\tilde{g}_t$ $\qquad\qquad\qquad\qquad$ $\diamond$ Update first-order gradient moment
7:     $\quad v_t = \beta_2 v_{t-1} + (1 - \beta_2)\tilde{g}_t^2$ $\qquad\qquad\qquad\qquad$ $\diamond$ Update second-order gradient moment
8:     $\quad \theta_{t+1} = (1 - \eta_t \lambda)\theta_t - \eta \frac{m_t}{\sqrt{v_t} + \epsilon}$ $\qquad\quad$ $\diamond$ Update the model parameters with weight decay
9: **end for**

---

**Lemma 8.** *Under Assumptions 1-5, for all iterates of Algorithm 4 we have*

$$\begin{aligned}
\mathbb{E}\left[\|m_t - \nabla f(\theta_t)\|^2\right] \leq &\left(1 - \frac{\beta_1}{2}\right) \mathbb{E}\left[\|m_{t-1} - \nabla f(\theta_{t-1})\|^2\right] + \frac{2}{\beta_1} L^2 \mathbb{E}\left[\|\theta_t - \theta_{t-1}\|^2\right] \\
&+ \beta_1 \mathbb{E}\|\nabla f(\theta_t) - \tilde{g}_t\|^2.
\end{aligned}$$

*Proof.* We start our proof from

$$\mathbb{E}\left[\|m_t - \nabla f(\theta_t)\|^2\right] = \mathbb{E}\left[\|(1-\beta_1)m_{t-1} + \beta_1\tilde{g}_t - \nabla f(\theta_t)\|^2\right]$$
$$= \mathbb{E}\left[\|(1-\beta_1)m_{t-1} + (1-\beta_1)\nabla f(\theta_{t-1})\right.$$
$$\left. - (1-\beta_1)\nabla f(\theta_{t-1}) + \beta_1\tilde{g}_t - \nabla f(\theta_t)\|^2\right]$$
$$= \mathbb{E}\left[\|(1-\beta_1)m_{t-1} + (1-\beta_1)\nabla f(\theta_{t-1}) - (1-\beta_1)\nabla f(\theta_t)\right.$$
$$\left. - (1-\beta_1)\nabla f(\theta_{t-1}) + \beta_1\tilde{g}_t - \beta_1\nabla f(\theta_t)\|^2\right].$$

Using Jensen's inequality we have

$$\mathbb{E}\left[\|m_t - \nabla f(\theta_t)\|^2\right] \leq (1-\beta_1)\mathbb{E}\left[\|m_{t-1} - \nabla f(\theta_{t-1}) + \nabla f(\theta_{t-1}) - \nabla f(\theta_t)\|^2\right]$$
$$+ \beta_1\mathbb{E}\left[\|\tilde{g}_t - \nabla f(\theta_t)\|^2\right].$$

Using Young's inequality we have

$$\mathbb{E}\left[\|m_t - \nabla f(\theta_t)\|^2\right] \leq (1-\beta_1)(1+b)\mathbb{E}\left[\|m_{t-1} - \nabla f(\theta_{t-1})\|^2\right]$$
$$+ (1-\beta_1)\left(1 + \frac{1}{b}\right)\mathbb{E}\|\nabla f(\theta_{t-1}) - f(\theta_t)\|^2$$
$$+ \beta_1\mathbb{E}\|\nabla f(\theta_t) - \tilde{g}_t\|^2.$$

Setting $b = \frac{\beta_1}{2}$ we have $(1-\beta_1)\left(1 + \frac{\beta_1}{2}\right) \leq 1 - \frac{\beta_1}{2}$ and $(1-\beta_1)\left(1 + \frac{2}{\beta_1}\right) \leq \frac{2}{\beta_1}$:

$$\mathbb{E}\left[\|m_t - \nabla f(\theta_t)\|^2\right] \leq \left(1 - \frac{\beta_1}{2}\right)\mathbb{E}\left[\|m_{t-1} - \nabla f(\theta_{t-1})\|^2\right]$$
$$+ \frac{2}{\beta_1}\mathbb{E}\|\nabla f(\theta_{t-1}) - f(\theta_t)\|^2 + \beta_1\mathbb{E}\|\nabla f(\theta_t) - \tilde{g}_t\|^2.$$

Combining this bound with $L$-smoothness we obtain

$$\mathbb{E}\left[\|m_t - \nabla f(\theta_t)\|^2\right] \leq \left(1 - \frac{\beta_1}{2}\right)\mathbb{E}\left[\|m_{t-1} - \nabla f(\theta_{t-1})\|^2\right]$$
$$+ \frac{2}{\beta_1}L^2\mathbb{E}\|\theta_{t-1} - \theta_t\|^2 + \beta_1\mathbb{E}\|\nabla f(\theta_t) - \tilde{g}_t\|^2.$$

$\square$

**Lemma 9.** *Under Assumptions 1-5, for all iterates of Algorithm 4 we have*
$$\mathbb{E}\left[\|\nabla f(\theta_t) - \tilde{g}_t\|^2\right] \leq 9\mathbb{E}\left[\|e_t\|^2\right] + 6G^2 + 3\sigma^2.$$

*Proof.* We start from

$$\mathbb{E}\left[\|\nabla f(\theta_t) - \tilde{g}_t\|^2\right] = \mathbb{E}\left[\|\mathcal{C}(e_t + g_t) - \nabla f(\theta_t)\|^2\right]$$
$$= \mathbb{E}\left[\|\mathcal{C}(e_t + g_t) - (e_t + g_t) + (e_t + g_t) - \nabla f(\theta_t)\|^2\right]$$
$$\leq 3\mathbb{E}\left[\|\mathcal{C}(e_t + g_t) - (e_t + g_t)\|^2\right]$$
$$+ 3\mathbb{E}\left[\|e_t\|^2\right] + 3\mathbb{E}\left[\|\nabla f(\theta_t) - g_t\|^2\right].$$

Using definition of contractive compressor we have

$$\mathbb{E}\left[\|\nabla f(\theta_t) - \tilde{g}_t\|^2\right] \leq 3q^2\mathbb{E}\left[\|e_t + g_t\|^2\right]$$
$$+ 3\mathbb{E}\left[\|e_t\|^2\right] + 3\mathbb{E}\left[\|\nabla f(\theta_t) - g_t\|^2\right].$$

Using Young's inequality we have

$$\mathbb{E}\left[\|\nabla f(\theta_t) - \tilde{g}_t\|^2\right] \leq 6q^2\mathbb{E}\left[\|e_t\|^2\right] + 6q^2\mathbb{E}\left[\|g_t\|^2\right]$$
$$+ 3\mathbb{E}\left[\|e_t\|^2\right] + 3\mathbb{E}\left[\|\nabla f(\theta_t) - g_t\|^2\right]$$
$$\leq 9\mathbb{E}\left[\|e_t\|^2\right] + 6G^2 + 3\sigma^2.$$

$\square$

**Lemma 10.** *Under Assumptions 1-5, for all iterates of Algorithm 4 we have*

$$\mathbb{E}\left[\|e_t\|^2\right] \leq \frac{(1+\omega)^2 q^2}{\left(1-(1+\omega)q\right)^2} G^2$$

*Proof.* Using definition of contractive compressor we have

$$\mathbb{E}\left[\|e_{t+1}\|^2\right] = \mathbb{E}\left[\|\mathcal{Q}\left(e_t + g_t - \mathcal{C}(e_t + g_t)\right)\|^2\right]$$

$$\leq (1+\omega)^2 q^2 \mathbb{E}\left[\|e_t + g_t - \mathcal{C}(e_t + g_t)\|^2\right].$$

Using Young's inequality we have

$$\mathbb{E}\left[\|e_{t+1}\|^2\right] \leq (1+\omega)^2 q^2 \left(1+a\right)\mathbb{E}\left[\|e_t\|^2\right] + (1+\omega)^2 q^2 \left(1+\frac{1}{a}\right)\mathbb{E}\left[\|g_t\|^2\right].$$

We need to satisfy the following condition:

$$(1+\omega)^2 q^2 (1+a) \leq (1+\omega)q.$$

It holds for $0 \leq \omega$, $0 \leq q < 1$, $(1+\omega)q < 1$ and $a = \frac{1}{(1+\omega)q} - 1$. Using this parameters we have

$$\mathbb{E}\left[\|e_{t+1}\|^2\right] \leq (1+\omega)q\mathbb{E}\left[\|e_t\|^2\right] + (1+\omega)^2 q^2 \left(\frac{1}{1-(1+\omega)q}\right)\|g_t\|^2$$

$$\leq (1+\omega)q\mathbb{E}\left[\|e_t\|^2\right] + \frac{(1+\omega)^2 q^2}{1-(1+\omega)q}\mathbb{E}\left[\|g_t\|^2\right].$$

Unrolling this recursion allows us to obtain

$$\mathbb{E}\left[\|e_{t+1}\|^2\right] \leq \left((1+\omega)q\right)^t \mathbb{E}\left[\|e_1\|^2\right] + \sum_{t=1}^{T}\left((1+\omega)q\right)^t \frac{(1+\omega)^2 q^2}{1-(1+\omega)q}G^2$$

$$\leq \left((1+\omega)q\right)^t \mathbb{E}\left[\|e_1\|^2\right] + \frac{1}{1-(1+\omega)q}\frac{(1+\omega)^2 q^2}{1-(1+\omega)q}G^2$$

$$\leq \frac{(1+\omega)^2 q^2}{\left(1-(1+\omega)q\right)^2}G^2,$$

last inequality holds because $e_1 = 0$. □

**Lemma 11.** *Under Assumptions 1-5, for all iterates of Algorithm 4 we have*

$$\mathbb{E}\left[\|\tilde{g}_t\|^2\right] \leq 4(1+q^2)\left(1+\frac{(1+\omega)^2 q^2}{(1-(1+\omega)q)^2}\right)G^2.$$

*Proof.*

$$\mathbb{E}\left[\|\tilde{g}_t\|^2\right] = \mathbb{E}\left[\|\mathcal{C}\left(g_t + e_t\right)\|^2\right]$$

$$= \mathbb{E}\left[\|\mathcal{C}\left(g_t + e_t\right) - \left(g_t + e_t\right) + \left(g_t + e_t\right)\|^2\right]$$

$$\leq 2\mathbb{E}\left[\|\mathcal{C}\left(g_t + e_t\right) - \left(g_t + e_t\right)\|^2\right] + 2\mathbb{E}\left[\|g_t + e_t\|^2\right]$$

$$\leq 2(1+q^2)\mathbb{E}\left[\|g_t + e_t\|^2\right].$$

Using Lemma 10 we obtain

$$\mathbb{E}\left[\|\tilde{g}_t\|^2\right] \leq 4(1+q^2)\left(1+\frac{(1+\omega)^2 q^2}{(1-(1+\omega)q)^2}\right)G^2.$$

□

**Lemma 12** (From paper: Zhou et al. [2024b]). *Let us consider the update rule:*

$$\theta_{t+1} = (1 - \eta\lambda)\theta_t - \eta_t \frac{m_t}{\sqrt{v_t + \varepsilon}}.$$

*For brevity, we denote $\widehat{v}_t = \sqrt{v_t + \varepsilon}$. Also we define*

$$u_t := m_t + \lambda\theta_t \otimes \widehat{v}_t,$$

*where $\otimes$ denotes element-wise product. Moreover, we also define $\widetilde{f}(\theta_t)$ as follows:*

$$\widetilde{f}(\theta_t) = f(\theta_t) + \lambda_t \|\theta_t\|_{\widehat{v}_t}^2$$

*where $\lambda_t = \frac{\lambda}{2}\sum_{i=1}^{t}\left(\frac{1-q}{2}\right)^i$ for $t > 0$, $\lambda_0 = 0$ with $0 < q < 1$ and $\|\theta_t\|_{\hat{v}_t} = \sqrt{\langle\theta_t, \hat{v}_t \otimes \theta_t\rangle}$. Also let $c_1 \leq \|\widehat{v}_t\|_\infty < c_2$, then iterates of Algorithm 4 satisfy*

$$\widetilde{f}(\theta_t) \leq \widetilde{f}(\theta_{t-1}) + \frac{\eta_t}{2c_1}\|\nabla f(\theta_{t-1}) - m_{t-1}\|^2$$

$$- \frac{\eta_t}{2c_2}\left\|\nabla\widetilde{f}(\theta_{t-1})\right\|^2 - \frac{\eta_t}{4c_2}\|u_{t-1}\|^2.$$

**Lemma 13** (From paper: Zhou et al. [2024b]). *Assume that $c_{s,\infty} \leq \|\tilde{g}_t\|_\infty \leq c_\infty$, then we have*

$$\|m_t\|_\infty \leq c_\infty, \quad \|v_t + \epsilon\|_\infty c_\infty^2 + \epsilon, \quad \left\|\frac{(v_t + \epsilon)^p}{(v_{t+1} + \epsilon)^p}\right\|_\infty \in [1 - \mu, 1 + \mu]\,(\forall p \in (0, 1)),$$

*where $\mu = \frac{\beta_2 c_\infty^2}{c_{s,\infty}^2 + \epsilon}$.*

**Theorem 5.** *Let Assumptions 1 to 5 hold. Define $\Psi_t = \widetilde{f}(\theta_t) + \frac{\eta_t}{2c_1\beta_1}\mathbb{E}\left[\|m_t - \nabla f(\theta_t)\|^2\right]$. With $\eta_t = \eta \leq \frac{\beta_1 c_1}{2L}\sqrt{\frac{c_1}{2c_2}}$, Algorithm 4 satisfies*

$$\frac{1}{T}\sum_{t=1}^{T}\left\|\nabla\widetilde{f}(\theta_{t-1})\right\|^2 \leq \frac{2c_2}{\eta T}\Psi^0$$

$$+ \frac{c_2}{c_1}\left(9\left(1 + \frac{(1+\omega)^2 q^2}{(1 - (1+\omega)q)^2}\right)G^2 + 6G^2 + 3\sigma^2\right).$$

*Proof.* We start from main lemma and lemma for momentum, summing inequalities together we obtain

$$\widetilde{f}(\theta_t) + V\mathbb{E}\left[\|m_t - \nabla f(\theta_t)\|^2\right] \leq \widetilde{f}(\theta_{t-1}) + \frac{\eta_t}{2c_1}\|\nabla f(\theta_{t-1}) - m_{t-1}\|^2$$

$$- \frac{\eta_t}{2c_2}\left\|\nabla\widetilde{f}(\theta_{t-1})\right\|^2 - \frac{\eta_t}{4c_2}\|u_{t-1}\|^2$$

$$+ V\left(1 - \frac{\beta_1}{2}\right)\mathbb{E}\left[\|m_{t-1} - \nabla f(\theta_{t-1})\|^2\right]$$

$$+ V\frac{2}{\beta_1}L^2\mathbb{E}\|\theta_{t-1} - \theta_t\|^2 + V\beta_1\mathbb{E}\|\nabla f(\theta_t) - \tilde{g}_t\|^2.$$

Using previous lemmas we have

$$\widetilde{f}(\theta_t) + V\mathbb{E}\left[\|m_t - \nabla f(\theta_t)\|^2\right] \leq \widetilde{f}(\theta_{t-1}) + \frac{\eta_t}{2c_1}\|\nabla f(\theta_{t-1}) - m_{t-1}\|^2$$

$$- \frac{\eta_t}{2c_2}\left\|\nabla\widetilde{f}(\theta_{t-1})\right\|^2 - \frac{\eta_t}{4c_2}\|u_{t-1}\|^2$$

$$+ V\left(1 - \frac{\beta_1}{2}\right)\mathbb{E}\left[\|m_{t-1} - \nabla f(\theta_{t-1})\|^2\right]$$

$$+ V\frac{2}{\beta_1}L^2\mathbb{E}\|\theta_{t-1} - \theta_t\|^2$$

$$+ V\beta_1\left(9\left(1 + \frac{(1+\omega)^2 q^2}{(1 - (1+\omega)q)^2}\right)G^2 + 6G^2 + 3\sigma^2\right).$$

Using $\theta_t - \theta_{t-1} = -\eta_t \frac{u_{t-1}}{\widehat{v}_{t-1}}$ we have

$$\widetilde{f}(\theta_t) + V\mathbb{E}\left[\|m_t - \nabla f(\theta_t)\|^2\right] \leq \widetilde{f}(\theta_{t-1}) - \left(\frac{\eta_t}{4c_2} - \frac{2VL^2\eta_t^2}{\beta_1 c_1^2}\right)\|u_{t-1}\|^2$$
$$- \frac{\eta_t}{2c_2}\left\|\nabla\widetilde{f}(\theta_{t-1})\right\|^2$$
$$+ \left(V\left(1 - \frac{\beta_1}{2}\right) + \frac{\eta_t}{2c_1}\right)\mathbb{E}\left[\|m_{t-1} - \nabla f(\theta_{t-1})\|^2\right]$$
$$+ V\beta_1\left(9\left(1 + \frac{(1+\omega)^2 q^2}{(1 - (1+\omega)q)^2}\right)G^2 + 6G^2 + 3\sigma^2\right).$$

Using $V = \frac{\eta_t}{2c_1\beta_1}$ and $\Psi_t = \widetilde{f}(\theta_t) + \frac{\eta_t}{2c_1\beta_1}\mathbb{E}\left[\|m_t - \nabla f(\theta_t)\|^2\right]$ we have

$$\Psi_t \leq \Psi_{t-1} - \frac{\eta_t}{4c_2}\left(1 - \frac{4c_2 L^2\eta_t^2}{\beta_1^2 c_1^3}\right)\|u_{t-1}\|^2$$
$$- \frac{\eta_t}{2c_2}\left\|\nabla\widetilde{f}(\theta_{t-1})\right\|^2$$
$$+ \frac{\eta_t}{2c_1}\left(9\left(1 + \frac{(1+\omega)^2 q^2}{(1 - (1+\omega)q)^2}\right)G^2 + 6G^2 + 3\sigma^2\right).$$

Using $\eta_t = \eta \leq \frac{\beta_1 c_1}{2L}\sqrt{\frac{c_1}{2c_2}}$ we have

$$\Psi_t \leq \Psi_{t-1} - \frac{\eta_t}{2c_2}\left\|\nabla\widetilde{f}(\theta_{t-1})\right\|^2$$
$$+ \frac{\eta}{2c_1}\left(9\left(1 + \frac{(1+\omega)^2 q^2}{(1 - (1+\omega)q)^2}\right)G^2 + 6G^2 + 3\sigma^2\right).$$

Summing from $t = 1$ to $T$ we have

$$\frac{1}{T}\sum_{t=1}^{T}\left\|\nabla\widetilde{f}(\theta_{t-1})\right\|^2 \leq \frac{2c_2}{\eta T}\Psi^0 + \frac{c_2}{c_1}\left(9\left(1 + \frac{(1+\omega)^2 q^2}{(1 - (1+\omega)q)^2}\right)G^2 + 6G^2 + 3\sigma^2\right).$$

$\square$

**Discussion.** This result is similar to the one from Zhou et al. [2024b] in the non-convex case, where the decay rate for the first term is $\mathcal{O}\left(\frac{1}{T}\right)$ and the second term is a non-vanishing $\mathcal{O}\left(\beta_1\frac{c_2}{c_1}\sigma^2\right)$. In our result, the non-vanishing term is proportional to $\mathcal{O}\left(\frac{c_2}{c_1}\left(\sigma^2 + G^2\right)\right)$.

A key difference is that our term is not proportional to $\beta_1$. It is important to note that $\beta_1$ typically takes a value close to 1 in practical applications, meaning its influence on the bound is minimal. Therefore, even though our result does not directly involve $\beta_1$, the impact on the overall bound is not significantly different.

Moreover, our bound includes an additional term proportional to $G^2$, which represents the gradient norm squared. This makes the bound slightly worse compared to the result in Zhou et al. [2024b]. However, this degradation is only by a constant factor, which means that while the theoretical bound may be worse, the practical implications are often negligible.

In summary, our result aligns closely with previous findings, with differences primarily in the constant factors and the presence of $G^2$. Despite these differences, the practical performance remains largely unaffected, ensuring that the bound remains robust and applicable in a variety of scenarios.

## F  Error Feedback applied to GaLore

### F.1  Behaviour of the Error Feedback Mechanism

The GaLore low-rank updates introduced by [Zhao et al., 2024] enable the compression of optimizer states by performing learning updates on a lower-dimensional subspace. In this approach, the

optimizer receives gradients projected on a defined learning subspace. Theoretical convergence guarantees are provided under a "stable rank" assumption, where learning subspace is fixed during training. However, in practice, convergence is attained by occasionally updating the learning subspace and allowing full space learning to better align with the gradient trajectory during training.

Here, it is useful to draw an analogy with the TopK method, as the occasional updates of the learning subspace resembles working with a fixed mask for many steps. Using a fixed mask would result in discarding the same coordinates of the gradient at each step. Similarly, in the case of low-rank updates, components orthogonal to the same learning subspace are discarded at each step.

The systematic nature of the information discarded by compression carries significant implications for error feedback behavior. Over multiple steps, the error accumulates gradient components belonging to the orthogonal space of the same learning subspace. Consequently, by linearity, the error itself resides in the orthogonal space of this learning subspace. As a result, when the error is passed to the accumulator, its projection onto the learning space is effectively disregarded until it is potentially utilized at the specific step when the learning subspace is updated. Therefore, the behavior of error feedback in the case of low-rank updates is non-standard: it accumulates gradient components over numerous steps before unloading them all at once.

For a better understanding, we derive analytical evidence for the described behaviour by induction. Let $L$ be fixed learning subspace and assume that $e_{t-1} \in L^\perp$. Then, gradient passed to the optimizer at step $t$ is: $\mathcal{C}_{GaLore}(a_t) = proj_L(a_t) = proj_L(e_{t-1} + g_t) = proj_L(g_t)$ where error feedback is discarded. Thus, $e_t = a_t - \mathcal{C}_{GaLore}(a_t) = e_{t-1} + g_t - proj_L(g_t) = e_{t-1} + proj_{L^\perp}(g_t) \in L^\perp$ which completes the induction.

Assume now that learning subspace is updated every $T$ steps, and denote $L_t$ the learning subspace at step $t$. Then, a similar induction leads to:

$$e_t = \sum_{i=1}^{t} \overset{t}{\underset{j=i}{\circ}} proj_{L_j^\perp}(g_i) = \sum_{i=1}^{t} \overset{\lfloor \frac{t}{T} \rfloor}{\underset{j=\lfloor \frac{i}{T} \rfloor}{\circ}} proj_{L_{jT}^\perp}(g_i)$$

$$a_{kT} = proj_{L_{kT}}(g_{kT} + e_{kT-1}) = proj_{L_{kT}}(g_{kT}) + \sum_{i=1}^{kT-1} proj_{L_{kT}} \circ (\overset{t}{\underset{j=i}{\circ}} proj_{L_j^\perp})(g_i)$$

### F.2 Consequences on Training

Such behaviour of the error feedback mechanism results in the dominance of the error norm over the gradient norm. Before learning subspace updates, the error is the sum over past gradient components that belong to the orthogonal of the current learning subspaces. Since these components represent descent directions that were not used, they are not expected to compensate each other on average. Consequently, between learning subspace updates, the error norm is expected to grow linearly. Figure 8 provides evidence of such linear growth of the error norm during fine-tuning of RoBERTa-base model on GLUE/MNLI task.

It implies that known analysis techniques [Alistarh et al., 2018, Karimireddy et al., 2019] of convergence for the error feedback mechanism do not apply to GaLore. Indeed, such proofs rely on the assumption that the compression operator is contractive, as it allows the error to be bounded. Given a fixed vector, low-rank compression based on its singular value decomposition is a contraction operator. However, in our case, the compression is based on a previously-computed singular value decomposition and therefore may not be a contraction operator for newly computed gradients. The extreme case being when the gradient is orthogonal to the learning subspace, in which case the compression operator returns the null vector. Figure 8 shows that during training the error norm is not on the same order of magnitude of the gradient norm.

The dominance of the error over the gradient also has effects on space exploration, as the learning subspaces are computed from the singular value decomposition of the accumulator (i.e. the sum of the gradient and the error). Since the main components of the accumulator belong to the orthogonal of current learning subspaces, successive learning subspaces will tend to be orthogonal to each other. This allows errors to be effectively passed to the optimizer, but all at once which can introduce irregularities in the learning trajectory. However, it also implies that learning is performed on a learning subspace that is suboptimal in terms of the direction of the gradient, but this may help

Figure 8: Dynamics of the norm of the error compared to norm of the gradient (of output of the 3rd attention layer) during fine-tuning of RoBERTa-base model on GLUE/MNLI from surrogate GaLore with error feedback optimizer. We used hyperparameters from [Zhao et al., 2024], i.e. batch size 16, learning rate 0.00001, projection update gap 200, rank 4 and GaLore scale 4.

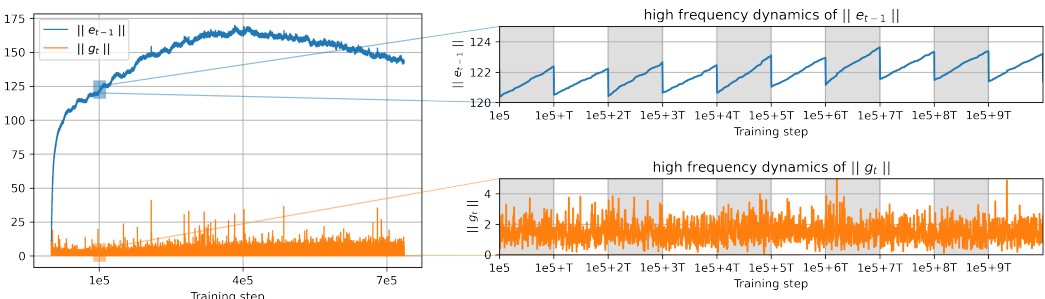

convergence by enforcing space exploration. See Figure 9 for examples of how induced orthogonality of successive learning subspaces affects the learning trajectory.

Figure 9: Optimization trajectory for Adam, GaLore-Adam and GaLore-Adam-EF for ill-conditioned function $f(x, y) = cos(\frac{5\pi}{4} x) + sin(\frac{7\pi}{4} y)$ starting from $(x_0, y_0) = (-\frac{1}{4}, \frac{1}{4})$ (on first row) and for Rosenbrock function starting from $(x_0, y_0) = (-\frac{1}{2}, 1)$ (on second row).

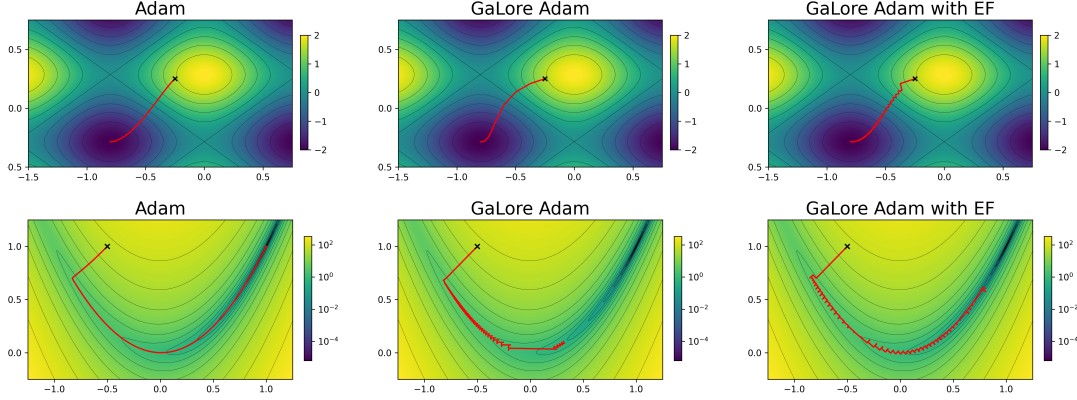

