# OpenReview forum: "MicroAdam: Accurate Adaptive Optimization with Low Space Overhead and Provable Convergence"
_NeurIPS.cc/2024/Conference — NeurIPS 2024 poster_

### Official Review · Reviewer_u3Hk · 2024-07-10

**Soundness:** 3
**Presentation:** 3
**Contribution:** 2
**Rating:** 6
**Confidence:** 3

**Summary:**

This paper proposes a new optimizer based on Adam, called MicroAdam, which reduces memory footprints, via sparisification, quantization, and error feedback. Theoretical convergence guarantees are provided. Empirical results show good performance on several fine-tuning tasks

**Strengths:**

1. This paper proposes a new optimizer based on Adam, called MicroAdam, which reduces memory footprints, via sparisification, quantization, and error feedback.
2. Theoretical convergence guarantees are provided.
3. Empirical results show good performance on several fine-tuning tasks
4. Efficient implementation on GPUs is provided.

**Weaknesses:**

1. The experiments are limitted to finetuning tasks. Note that GaLore also show good performance in pretraining tasks.
2. The theoretical analysis is actually based on AMSGrad instead of Adam.

---------------

My concerns are well addressed according to the author's feedback.

**Questions:**

1. In many cases, MicroAdam shows even better loss or accuracy than the original Adam, which doesn't really make sense since the compression methods should be lossy which typically incurs worse loss or accuracy. Is there any explanation of this phenomenon?

**Limitations:**

The limitations are well discussed.

---

> ### Author Rebuttal · Authors · 2024-08-06
>
> We would like to thank the reviewer for the feedback. We address your questions below.
>
> **Weakness 1:** Pretraining
>
> We would like to emphasize that we designed MicroAdam for the finetuning (FT) use case and our research question was “are all gradient entries useful for fine-tuning?”, which is why we test MicroAdam for FT tasks.
>
> We agree with the reviewer regarding MicroAdam’s convergence in Figure 4 and we can explain this behavior by emphasizing the high sparsity in MicroAdam’s update (which, in our experiments, is around 90%). However, our results show that we can achieve at least the same performance with only 45% memory compared to AdamW-8bit ($0.9d$ bytes for MicroAdam compared to $2d$ bites of AdamW-8bit).
>
> We address your concerns on effectiveness via two additional experiments:
>
> 1. To address the training from scratch / non-language tasks question, we trained ResNet-18 and ResNet-50 on ImageNet using MicroAdam and compared it with SGD, AdamW and AdamW-8bit. We provide results in the response PDF. Briefly, MicroAdam manages to obtain close to 72% accuracy, compared to 70% for SGD, AdamW and AdamW-8bit.
> - For ResNet-18, **MicroAdam uses ~10 MB** memory for the model state, compared to ~22 MB for AdamW-8bit ($2.2\times$ more), ~45 MB for SGD ($4.5\times$ more) and ~90 MB for AdamW ($9 \times$ more).
> - For ResNet-50, **MicroAdam uses ~22 MB** memory for the model state compared to ~49 MB for AdamW-8bit ($2.2 \times$ more), ~97 MB for SGD ($4.4 \times$ more) and ~195 MB for AdamW ($8.9 \times$ more).
>
> This is an indication that MicroAdam is effective for pretraining model for vision tasks.
>
> 2. To address the question of scale, in addition to existing experiments, we would like to emphasize the effectiveness of MicroAdam for SFT by training a Llama 2-13B model on the same GSM-8k dataset and comparing it against AdamW-8bit. We provide results in the response PDF. Briefly, our results show that MicroAdam can also recover the accuracy of AdamW-8bit for this task at a lower memory cost: **MicroAdam allows training on a single Nvidia H100 GPU with 80GB RAM**, the total running process requires 70 GB GPU memory (~59 GB for the model and activations and ~11 GB is used for the optimizer state), while AdamW-8bit (24.2 GB for the optimizer state) requires more than one 80GB GPU and need at least 2 GPUs. The need for more than one GPU for AdamW-8bit comes from the size of optimizer state, which is $2.2 \times$ larger.
>
> FT tasks require adapting a fraction of parameters (for example, LoRA) to be able to learn a downstream task. In contrast, pretraining (PT) requires updating all parameters in the model because there is a lot more knowledge to be embedded into the model compared to FT case and the dataset sizes are also much larger. In the paper we briefly explain that under the settings we choose for MicroAdam ($k=1\\%$ and $m=10$ gradients), the parameter update is at least 90% sparse, meaning that MicroAdam updates at most $10\\%$ of parameters in each layer at every optimization step.
>
> **Weakness 2:** Theoretical analysis based on AMSGrad
>
> The AMSGrad optimizer was introduced as a solution to a fundamental issue in Adam optimizer’s proof, where a state matrix was mistakenly supposed to be positive definite throughout the training (see Section 3 from Reddi et al. 2019 [1]). In Section 5 of [1], the authors provide a simple, one dimensional problem where Adam converges to the worst possible solution. Despite this clear evidence of unsatisfactory solution in simple settings, Adam (and the newer version AdamW [2]) is still the off-the-shelf optimizer when it comes to LLMs finetuning. We chose to build our theoretical analysis on the framework of AMSGrad since this algorithm fixes the above technical issue, and can therefore have a convergence proof (as opposed to Adam). Moreover, CompAMS introduces error feedback and provides analysis for AMSGrad in a distributed optimization setting. In this context, our work focuses on optimizing space usage, and introduces compression for error feedback in the CompAMS framework.
>
> **Question 1:** Explaining the better performance
>
> This behavior was also surprising for us. We believe that compression has a regularization effect by updating only 10% of the weights. Adding noise into training dynamics can both increase or decrease the loss or accuracy. It is not the case that lossy compression or noisy training must have higher loss or lower accuracy. For instance, if that were the case, then larger batch size (and implicitly lower noise in the stochastic gradient) should have implied better accuracy, which is not true as far as we are concerned. We are thinking about this in the context of full batch gradient descent, which does not generalize better than the mini-batch version.
>
> References:
> - [1] **On the convergence of Adam and beyond**, Reddi et al., 2019, available at https://arxiv.org/pdf/1904.09237
> - [2] **DECOUPLED WEIGHT DECAY REGULARIZATION**, Loshchilov and Hutter 2017, available at https://arxiv.org/pdf/1711.05101
> - [3] **ON DISTRIBUTED ADAPTIVE OPTIMIZATION WITH GRADIENT COMPRESSION**, Li et al., 2022, ICLR 2022, available at https://arxiv.org/pdf/2205.05632

---

> > ### Comment · Reviewer_u3Hk · 2024-08-12
> >
> > I thank the authors for the feedback. It seems that most of my concerns are addressed and I will raise the score.

---

### Official Review · Reviewer_C8Q3 · 2024-07-12

**Soundness:** 3
**Presentation:** 3
**Contribution:** 2
**Rating:** 6
**Confidence:** 2

**Summary:**

This paper proposes a memory-efficient Adam-based optimizer called MicroAdam. The key idea is to compress gradients using Top-K sparsification before passing them to the optimizer state, along with an error feedback vector that is itself compressed via quantization. For general smooth non-convex (and PL) functions, MicroAdam’s convergence rates are provided with constants depending on the choice of compressors. Experiments on fine-tuning tasks with BERT and LLaMA suggest that MicroAdam performs competitively with AdamW while using less memory.

**Strengths:**

- This work provides theoretical convergence guarantees under commonly used assumptions. Previous work on memory-efficient Adam often relied more on heuristics.
- Performance on fine-tuning tasks is comparable to uncompressed AdamW, with lower memory cost.

**Weaknesses:**

- Both theoretical and practical performance, as well as memory savings, depend heavily on the choice of compressors. These compressors can be complex to implement and require specific techniques to avoid memory overhead on GPUs. The difficulty and complex details of implementation might hinder impact as it is not easy to incorporate in practice.
- The approach is limited to fine-tuning tasks, and its effectiveness for LLM pre-training or non-language tasks remains unclear.

**Questions:**

The condition $(1+\omega)q \leq 1$ is necessary for theoretical convergence. Do the compressors used in the experiment section satisfy this requirement? Generally, how do the authors suggest finding appropriate compressors for unfamiliar tasks?

**Limitations:**

Limitations are well-addressed

---

> ### Author Rebuttal · Authors · 2024-08-06
>
> We would like to thank the reviewer for the feedback. We address your questions below.
>
> **Weakness 1:** Complexity of compressors
>
> We agree that memory savings depend on the choice of compressors, and that efficient implementations are needed. However, we do have the advantage that both gradient sparsification, quantization, and low-rank compression are well-understood from distributed optimization, and one can build on prior work in the area, as well as good support in popular frameworks such as Pytorch. Specifically, the implementation we provide is not very complex, and can be further optimized along the lines we sketched in the response to **Weaknesses 4 and 6** of **Reviewer nVyV**.
>
> **Weakness 2:** Pretraining
>
> We would like to emphasize that we designed MicroAdam for the finetuning (FT) use case and our research question was “are all gradient entries useful for fine-tuning?”, which is why we test MicroAdam for FT tasks.
>
>
> We agree with the reviewer regarding MicroAdam’s convergence in Figure 4 and we can explain this behavior by emphasizing the high sparsity in MicroAdam’s update (which, in our experiments, is around 90%). However, our results show that we can achieve at least the same performance with only 45% memory compared to AdamW-8bit ($0.9d$ bytes for MicroAdam compared to $2d$ bites of AdamW-8bit).
>
> We address your concerns on effectiveness via two additional experiments:
>
> 1. To address the training from scratch / non-language tasks question, we trained ResNet-18 and ResNet-50 on ImageNet using MicroAdam and compared it with SGD, AdamW and AdamW-8bit. We provide results in the response PDF. Briefly, MicroAdam manages to obtain close to 72% accuracy, compared to 70% for SGD, AdamW and AdamW-8bit.
> - For ResNet-18, **MicroAdam uses ~10 MB** memory for the model state, compared to ~22 MB for AdamW-8bit ($2.2\times$ more), ~45 MB for SGD ($4.5\times$ more) and ~90 MB for AdamW ($9 \times$ more).
> - For ResNet-50, **MicroAdam uses ~22 MB** memory for the model state compared to ~49 MB for AdamW-8bit ($2.2 \times$ more), ~97 MB for SGD ($4.4 \times$ more) and ~195 MB for AdamW ($8.9 \times$ more).
>
> This is an indication that MicroAdam is effective for pretraining model for vision tasks.
>
> 2. To address the question of scale, in addition to existing experiments, we would like to emphasize the effectiveness of MicroAdam for SFT by training a Llama 2-13B model on the same GSM-8k dataset and comparing it against AdamW-8bit. We provide results in the response PDF. Briefly, our results show that MicroAdam can also recover the accuracy of AdamW-8bit for this task at a lower memory cost: **MicroAdam allows training on a single Nvidia H100 GPU with 80GB RAM**, the total running process requires 70 GB GPU memory (~59 GB for the model and activations and ~11 GB is used for the optimizer state), while AdamW-8bit (24.2 GB for the optimizer state) requires more than one 80GB GPU and need at least 2 GPUs. The need for more than one GPU for AdamW-8bit comes from the size of optimizer state, which is $2.2 \times$ larger.
>
> FT tasks require adapting a fraction of parameters (for example, LoRA) to be able to learn a downstream task. In contrast, pretraining (PT) requires updating all parameters in the model because there is a lot more knowledge to be embedded into the model compared to FT case and the dataset sizes are also much larger. In the paper we briefly explain that under the settings we choose for MicroAdam ($k=1\\%$ and $m=10$ gradients), the parameter update is at least 90% sparse, meaning that MicroAdam updates at most $10\\%$ of parameters in each layer at every optimization step.
>
> **Question 1:** Compressors' properties
>
> The condition $(1+\omega) q < 1$ is theoretical and parameters $q$ and $\omega$ are worst case bounds. In the experiments we apply much more compression than the theoretical worst case condition allows us. In general, Top-K, low-rank and quantization are the main compression operators widely studied theoretically and used in practice.

---

> ### Comment · Reviewer_C8Q3 · 2024-08-14
> **Response to authors**
>
> Thank you for addressing my concerns and conducting additional vision pretraining experiments. The accuracy of AdamW seems a bit lower than usual. I don't think one can conclude that MicroAdam achieves better accuracy than AdamW in this case without further fine-tuning. However, I believe it sufficiently demonstrates MicroAdam's behavior in pretraining, so I will raise my score.

---

### Official Review · Reviewer_nVyV · 2024-07-12

**Soundness:** 3
**Presentation:** 3
**Contribution:** 2
**Rating:** 4
**Confidence:** 4

**Summary:**

The paper introduces MicroAdam, a novel optimizer designed to improve memory efficiency while maintaining competitive performance with established optimizers such as Adam. The authors provide theoretical analyses and experimental results to demonstrate the benefits of MicroAdam in various settings.

**Strengths:**

- **Theoretical Analysis**: The paper includes comprehensive theoretical analyses of memory consumption and the guarantee of convergence.
- **Experimental Validation**: The experimental results showcase the potential of MicroAdam in reducing memory consumption, with detailed comparisons to existing methods.

**Weaknesses:**

- **Motivation in Figure 1**: Figure 1 does not effectively provide additional motivation for MicroAdam, as it represents a 2-dimensional non-stochastic problem with only choosing k=1 for top-k compression. This toy example can be solved by many heuristic methods or parameter adjustments, which diminishes the unique motivation for MicroAdam.

- **Convergence Speed**: The convergence speed of MicroAdam is asymptotically similar to AMSGrad rather than Adam itself. Given that the paper aims to improve Adam's memory footprint, it is unclear why the convergence speed matches that of AMSGrad, which theoretically has no significant speed difference from Adam's with respect to T.

- **Memory Consumption Analysis**: Sections 3.2 and C provide theoretical and simulated memory consumption analysis. However, the practical memory consumption during actual LLM training does not necessarily correlate strongly with the theoretical optimizer states' memory usage. For example, a small LLAMA model with sufficiently large batch size and token length can peak at over 70GB of memory usage (mainly due to activation memory), whereas the optimizer states' memory footprint might be under 100MB. Therefore, the paper should report more about real peak memory under appropriate settings rather than theoretical or simulated values, as the differences in optimizer states memory can be mitigated under certain settings and PyTorch's memory management mechanism.

- **Top-K Operator Memory Efficiency**: Despite the block-wise operation of the Top-K operator, if the gradients are in bf16 format, the memory overhead for 50% sparsity remains the same (due to the need to store 16-bit indices) or could even be higher when considering other memory overheads in the algorithm.

- **Lack of Training from Scratch Results**: A significant issue is the near absence of results for training from scratch, which is crucial for evaluating MicroAdam's effectiveness. For MicroAdam, training from scratch is more important. Table 1 shows that during fine-tuning, the training loss does not strictly correlate with the final metrics. For instance, GaLore has a much worse loss but maintains metrics comparable to those of other methods. This is why LLM fine-tuning metrics are often taken from the best of three epochs, not necessarily the last epoch, even if the training loss is the smallest in the final epoch. Additionally, Figure 4 indicates that MicroAdam's training efficiency is lower than baseline Adam, as MicroAdam's third epoch loss is nearly identical to Adam's second epoch loss.

- **Computational Efficiency**: MicroAdam appears to be less computationally efficient. Are there ways to improve its computational efficiency?

**Questions:**

See Weaknesses

**Limitations:**

See Weaknesses

---

> ### Author Rebuttal · Authors · 2024-08-06
>
> We would like to thank the reviewer for the feedback. We address your questions below.
>
> **Weakness 1:** About Figure 1
>
> The motivation for this illustration is to show that some form of EF is _necessary_ for good convergence, even in the case of toy instances. Specifically, prior heuristic methods, such as GaLoRE, perform gradient compression but simply _omit EF altogether_, and we wanted to show that this is infeasible in general. We agree with your point that various other heuristics could be applied, and we will clarify the motivation for this example or move it to the Appendix.
>
> In Figure 1 we show how error feedback (EF) fixes AdamW with Top-K compression. The plot on the left shows the optimization trajectory of the original Adam optimizer. The center plot illustrates the convergence of Top-K Adam when we only choose the largest coordinate from the accumulator (equivalent to 50% sparsity, since the problem is two-dimensional). In the end, on the right side we show that adding EF to Top-K Adam recovers the same optimization trajectory as the original Adam optimizer. Extrapolating to higher dimensional problems, our MicroAdam approach helps recover the trajectory of the original Adam optimizer, while using less memory.
>
> **Weakness 2:** Convergence speed
>
> The AMSGrad optimizer was introduced as a solution to a fundamental issue in Adam optimizer’s proof, where a state matrix was mistakenly supposed to be positive definite throughout the training (see Section 3 from Reddi et al. 2019 [1]). In Section 5 of [1], the authors provide a one dimensional problem where Adam converges to the worst possible solution. Despite this clear evidence of unsatisfactory solution in simple settings, Adam (and the newer version AdamW [2]) is still the off-the-shelf optimizer when it comes to LLMs finetuning. We chose to build our theoretical analysis on the framework of AMSGrad since this algorithm fixes the above technical issue, and can therefore have a convergence proof (as opposed to Adam). Moreover, CompAMS introduces EF and provides analysis for AMSGrad in a distributed optimization setting. In this context, our work focuses on space optimizations, and introduces compression for EF in the CompAMS framework while preserving convergence.
>
> **Weakness 3:** Memory consumption
>
> We agree with the example you gave in your comment and we would like to add a few details about our experimental work. We emphasize that the memory usage we reported in the paper was the maximal memory usage, read directly from the GPU. This means that a user must have at least that much memory available on the GPU to perform the experiment. Our memory usage is the overall memory used by the entire program, including model activation and gradient, as well as the batch of data used for forward pass and the optimizer state. In section 3.2 we provide the memory usage only for the optimizer state expressed in bytes ($0.9d$ for MicroAdam, $2d$ for AdamW-8bit and $4d$ for AdamW) and this is the same for any model with $d$ parameters. Concretely, for Llama-2-7b, we have 25.1GB for AdamW, 12.5GB for AdamW-8bit and **5.65 GB for MicroAdam**.
>
> In practical terms, our experiments consider settings where the memory footprint for orthogonal components (such as activation sizes) is minimized, such as  microbatch size 1 and global batch size 32 for Llama-2-7B. One can have lower memory usage with exactly the same model settings only if the optimizer state is lower and this is what we try to do with MicroAdam.
>
> **Weakness 4:** Top-K Memory Efficiency
>
> The idea of MicroAdam is to store highly sparse gradients in the buffer $\mathcal{G}$ and to compute the statistics $m_t$ and $v_t$ on the fly. In practice, we show that very high (at least 99% sparsity) can be induced in the gradients, and this allows us to have a much lower memory usage in practice. It is true that for moderate (50%) sparsity the savings would be minimal or negative, but one of our sources of novelty is in showing that much higher sparsity can be supported in practice, with respect to the information used to generate optimizer states. We can provide more details about our efficient implementation in the discussions after the rebuttal.
>
> **Weakness 5:** Pretraining
>
> Because of limited size for the number of characters in our response, we are kindly asking to read our comment about pretraining in our response to **Weakness 1** of **Reviewer u3Hk**.
>
> **Weaknesss 6:** Computational Efficiency
>
> To speed up the main operations at the core of MicroAdam, we already implemented CUDA kernels for quantization and to efficiently compute the statistics $m_t$ and $v_t$. However, we believe there are several ways to further improve computational efficiency. For instance, we can experiment with changing the number of thread blocks in CUDA kernels for the auxiliary operations, such as setting tensors to zero (line 6) and copying the top-k values from the accumulator $a_t$ at indices $\mathcal{I}_t$ to the matrix $\mathcal{V}$ that stores the values. At the moment, we use the maximum possible number of CUDA thread blocks to perform the operation, which might be improved in the context of layer-wise preconditioning. We did not prioritize such optimizations during the preparation of the paper because MicroAdam achieved its main goal, that of reducing memory usage by 55% (wrt AdamW-8bit) and 77.5% (wrt AdamW).
> We skip some details now due to limited sized response, but we are happy to explain more in the discussions phase.
>
> References:
> - [1] **On the convergence of Adam and beyond**, Reddi et al., 2019, available at https://arxiv.org/pdf/1904.09237
> - [2] **DECOUPLED WEIGHT DECAY REGULARIZATION**, Loshchilov and Hutter 2017, available at https://arxiv.org/pdf/1711.05101
> - [3] **ON DISTRIBUTED ADAPTIVE OPTIMIZATION WITH GRADIENT COMPRESSION**, Li et al., 2022, ICLR 2022, available at https://arxiv.org/pdf/2205.05632

---

### Official Review · Reviewer_8ZAo · 2024-07-12

**Soundness:** 3
**Presentation:** 2
**Contribution:** 3
**Rating:** 6
**Confidence:** 3

**Summary:**

The paper presents a new optimizer that approximates Adam but has a lower memory footprint. Adam stores for each parameter two additional values --- the exponential moving averages of the gradients and the gradient squared. The current optimizer saves space by using the fact that for most parameters, the gradient is quite small at each step, so could be ignored. The optimizer stores the most important gradients at each step, for several past steps. It also stores a low-resolution version of the gradient, which is added to the next gradient (the reason for this was not clear to me). The exponential moving averages for the gradients and gradients-squared are computed from the most important gradients stored at each step and are used in the Adam update.

The paper proves the convergence of the optimizer under some reasonable assumptions. The authors show that the method can be implemented efficiently on a GPU, and the performance is similar to Adam. The authors performed several experiments to show that their optimizer saves about 10-20% memory compared to Adam, while achieving similar accuracy.

**Strengths:**

The main idea is quite interesting - that most of the values of a gradient are small in magnitude, so need not be used to update the parameters --- only up to 10% of all the parameters are ever updated in any step, and these change only a little across steps.

The convergence results are a useful validation of the algorithm. The experimental results seem quite strong to me.

**Weaknesses:**

The presentation of the paper was not very clear to me. For example, k is not mentioned as one of the inputs in Algorithm 1. $\delta_1, \Delta_1$ are initialized to vectors in R^d, but then they seem to be scalars.  I also did not understand what the error correction was for.

**Questions:**

In your algorithm, why did you not use a_t or the current gradient in steps 9 and 10, since this is already present.

It looks like most of the memory is being used for other tasks, so reducing the memory by a factor of 4 resulted in a small overall reduction in memory.

---

> ### Author Rebuttal · Authors · 2024-08-06
>
> We would like to thank the reviewer for the feedback. We address your questions below.
>
> **About Summary:** Memory savings
>
> The memory savings of MicroAdam are ~20% if we compare the entire memory usage, but the key comparison point should be on the size of optimizer states, as this is the quantity we are minimizing. We restate the memory usages for optimizer states presented in section 3.2 of our paper for Llama-2-7B model:
> - AdamW: 25.10 GB
> - AdamW-8bit: 12.55 GB
> - MicroAdam: **_5.65 GB_**
>
> In our next revision, we will show the optimizer state in the tables containing results to clearly differentiate from the memory usage that is shared across all optimizers (e.g. model state, such as activations and gradient). As explained in section 3.2 of our paper, the memory footprint of MicroAdam is $0.9d$ bytes in any experiment, where $d$ is the model size, while AdamW and AdamW-8bit have $4d$ and $2d$, respectively. To express the memory savings in percentages, MicroAdam uses only 45% memory compared to AdamW-8bit ($0.9d$ vs $2d$) and 22.5% compared to AdamW ($0.9d$ vs $4d$).
>
> **Weakness 1:** Paper clarity
>
> **Algorithm 1:** We explain Algorithm 1 in detail in Section 3 of our paper. Please check the end of this answer where we extend the explanations to make sure our algorithm is well understood.
>
> **About $k$:** Indeed, $k$ is one of the inputs of the Algorithm 1 and  $\delta_1$, $\Delta_1$ should be initialized as scalars. We will fix these in the next revision.
>
> **About error correction:** Let us clarify the main steps of the proposed MicroAdam algorithm and address some of the questions you raised. As you noted, Adam stores two additional values for each parameter: the exponential moving average of the gradients ($m_t$) and the squared gradients ($v_t$). To avoid allocating twice the model size for $m_t$  and $v_t$, we compress the gradients using a sparsity-inducing operator Top-K, storing the most important gradient components at each step, for several past steps (these are stored in the buffer $\mathcal{G}$). However, we do not ignore the smaller gradients, but we store and accumulate them at 4-bit resolution, adding them to the next gradient to compute the model updates. Intuitively, instead of disregarding small gradient components in each iteration, we accumulate them and apply them in the next step. This ensures that all gradient information is eventually used to update the model parameters. Below we provide a detailed explanation of Algorithm 1.
>
> **Question 1:**
>
> The idea of our algorithm is to store 99% sparse gradients (e.g. only the largest 1% of values) into the buffer matrix $\mathcal{G}$. Then, this sparse gradient window is used to “materialize” the Adam statistics $m_t$ and $v_t$. This way, we make sure that we use only $0.9d$ bytes of memory instead of $2d$ as in AdamW-8bit or even $4d$ as in original AdamW. Below we provide a detailed explanation of Algorithm 1.
>
> **Question 2:**
>
> In all our experiments we report the memory usage for the entire process. We would like to emphasize that what should be compared is the size of optimizer states, since the settings for each experiment is the same and only the optimizer differs (e.g. batch size and model sizes are fixed). In section 3.2 we provide a detailed theoretical memory analysis for the optimizer states of AdamW (25.1GB), AdamW-8bit (12.5GB), **MicroAdam (5.65 GB)**. In the next revision we will clearly make a distinction between the memory usage for the optimizer state and the overall memory usage (which includes, among others, model activations and gradient). Please let us know if we misunderstood your question. We are happy to discuss and clarify everything about our work.
>
>
> **_Explanation for Algorithm 1_**: line $n$ refers to $n^{th}$ line in this algorithm:
>
> **Optimization step $t=1$:**
> In this case, the error feedback is completely zero as initialized in line 2
> - **Line 4**: the accumulator $a_t$ contains only the stochastic gradient
> - **Line 5**: if we suppose the accumulator $a_t$ is normally distributed with a mean of zero, then this line is equivalent to choosing $k=1\\%$ values from the tails (outliers) because we apply Top-K on the absolute values of the accumulator $a_t$. The Top-K operator returns indices of those outliers, which we store in $\mathcal{I}_t$, as well as the corresponding values $\mathcal{V}_t$ from $a_t$ **(not from $|a_t|$)**
> - **Line 6**: the outliers are erased from the accumulator $a_t$ because they will be transferred to the buffer matrix $\mathcal{G}$
> - **Line 7**: compute $\delta$ and $\Delta$ (the min and max values from the accumulator $a_t$) which will be used for the quantization at the next step.
> - **Line 8**: quantize the remaining values in the accumulator $a_t$ **without the outliers in the tails** using procedure $Q$ in Alg. 2. **This error will be dequantized and added to the stochastic gradient at the next optimization step $t+1$**. This is the point in our algorithm when we preserve the error instead of discarding it.
> - **Line 9**: add the new set of outliers to the ring buffer $\mathcal{G}$.
> - **Lines 10 and 11**: compute the first and second order moment statistics $\hat{m_t}$ and $\hat{v_t}$ over the ring buffer $\mathcal{G}$ using Alg. 3.
> - **Line 12**: update the model parameters as usually done in Adam
>
> **Optimization step $t \geq 2$:**
> The only change compared to optimization step $t=1$ is that the error feedback $e_t$ is not zero anymore, but will contain the quantized values not selected by Top-K (inliers) and will be added to the gradient. After that, the algorithm works exactly as explained above, starting with line 5.

---

### Official Review · Reviewer_2Efz · 2024-07-15

**Soundness:** 2
**Presentation:** 1
**Contribution:** 1
**Rating:** 3
**Confidence:** 4

**Summary:**

The script proposed a memory-efficient method with a theoretical guarantee.

**Strengths:**

The topic on memory-efficient optimizers is important. I did not observe obvious flaws in the theory.

**Weaknesses:**

See below.

**Questions:**

See below.

**Limitations:**

**Major concern 1**: The contribution of the paper is rather weak. After reading the paper,  I still did not find what is new in this paper.  For instance:

**1-1: It is unclear what is new in the compression design of Algorithm 1.** I have a hard time figuring out the contribution of Algorithm 1.  Please summarize the significance and novelty of the proposed compression procedure in Algorithm 1. Please discuss how it is different from the existing ones and why do we need this new one. Please discuss the design principles and ideas.

 **1-2 It is unclear whether the proposed method is truly effective.**  There are much much more experiments are needed. Currently, only 7B SFT is provided, which is far from convincing. SFT is not enough to support the effectiveness of a new optimizer.  It seems unclear how the method would perform on LLM pre-training or non-language tasks.

 **1-3:  It is unclear what is new in the theory.**  Under the assumption of unbiased compression +  bounded variance + bounded gradient, this type of Algorithm 4 has already been extensively studied by the distributed community and so. Please highlight what is new and what is nontrivial, if there is any.





**Major concern 2: The presentation is quite poor.** For instance:

1. There is no explanation on the principle or idea behind the design of Algorithm 1.

2. It is good to see simple example like Figure 1. But unfortunately, nearly nothing is explained here. We can see EF helps  convergence, but we do not understand why. It seems just a toy example without any explanation on the insight.

3. The current logic flow of the script is confusing. It would be much easier to read if the authors:

   (i) introduce Algorithm 4 first (generic form of MicroAdam),

   (ii) provide the theoretical guarantee,

   (iii)then introduce Algorithm 1( a  specific form that you chose) ,

   (iv) discuss the implementation details,

   (v) then show  the experiments.

---

> ### Author Rebuttal · Authors · 2024-08-06
>
> We would like to thank the reviewer for the feedback. We address your questions below.
>
> **Limitation 1-1:** Contribution and novelty in MicroAdam
>
>
> We focus on reducing the memory cost of adaptive optimization, and start from the idea that not all gradient components carry useful information for optimization. Thus, we show that gradients can be sparsified **before used to compute the states in the Adam optimizer**. Intuitively, sparsity leads to significant savings in terms of the size of optimizer states.
> However, to ensure convergence both in theory and practice, it is necessary to incorporate an error correction mechanism **error feedback** (EF), made popular in distributed optimization. **Simply using EF does not lead to memory savings**, since the size of the error correction buffer is the same as the model size. At the same time, EF is critical for convergence, as also illustrated in Figure 1.
>
> Instead, **our key algorithmic innovation is in showing that the EF can itself be compressed** in the context of adaptive optimization, leading to significant memory savings while preserving convergence. The new parts of our algorithm, i.e. EF compression and decompression are highlighted **in blue** in Algorithm 1. Our practical contribution is in showing that sparsity in optimizer states can be efficiently leveraged for memory savings, at the scale of billion-parameter models.
>
> **Limitation 1-2:** Pretraining
>
> Because of limited size for the number of characters in our response, we are kindly asking to read our comment about pretraining in our response to **Weakness 1** of **Reviewer u3Hk**.
>
> **Limitation 1-3:** Novelty in the theory
>
> Motivated by the question of reducing storage cost, we are the first to consider **EF quantization** and to provide a convergence proof for it.
>
> The key point in the algorithm design, which allows us to obtain practical gains, is the ability to compress the EF accumulator through quantization. In the analytical view of MicroAdam, presented as Algorithm 4, the EF accumulator is denoted by $e_t$, and in line 5 this accumulator is updated and compressed via $e_{t+1} = Q(e_t + g_t - \tilde{g}_t)$.  As can be seen from examining our proof, quantizing the EF, although very simple algorithmically, significantly complicates the analysis. (Generally, we are not aware of any method that compresses the error accumulator and provides theoretical convergence guarantees in any setup. )
> Moreover, from the practical side, since the state we maintain are the quantized EF and the sparse gradients, our algorithm must re-compute optimizer states on-the-fly in every iteration based on this compressed state. Implementing this efficiently is a non-trivial challenge.
>
>
> **Major Concern-2:**
>
> **Algorithm 1**:
>
> We explain Algorithm 1 in detail in Section 3 of our paper. Please check the answer for **Reviewer 8ZAo** where we extend the explanations we already provided in section 3 of our paper to make sure our algorithm is well understood.
>
> **About Figure 1:**
>
> The motivation for this illustration is to show that some form of EF is **necessary** for good convergence, even in the case of toy instances. Specifically, prior heuristic methods, such as GaLoRE, perform gradient compression but simply **omit EF altogether**, and we wanted to show that this is infeasible in general. We agree with your point that various other heuristics could be applied, and we will clarify the motivation for this example or move it to the Appendix.
>
> In Figure 1 we show how EF fixes AdamW with Top-K compression. The left plot shows the optimization trajectory of the original Adam optimizer. The center plot shows the convergence of Top-K Adam when we only choose the largest coordinate from the accumulator (equivalent to 50% sparsity, since the problem is two-dimensional). In the right side we show that adding EF to Top-K Adam recovers the same optimization trajectory as the original Adam optimizer. Extrapolating to higher dimensional problems, our MicroAdam approach helps recover the trajectory of the original Adam optimizer, while using less memory.
>
> **Major Concern-3:** Sections order in the paper
>
> Thank you for this suggestion; we will re-order the presentation to introduce the general algorithm, followed by the implementable version.

---

> ### Comment · Reviewer_2Efz · 2024-08-13
> **I will keep my score**
>
> Thanks a lot for the response.
>
> I still think it is rather weak to validate the efficacy of a new optimizer by  SFT on Llama.  Further, I kindly disagree with authors comment that "our research question was "are all gradient entries useful for fine-tuning?” " This is not shown in the script. The whole script is written in the style of proposing a new optimizer for generic training procedures (including pre-training), not just for SFT.
>
> I will keep my score.

---

> > ### Author Response · Authors · 2024-08-13
> > **Author Response**
> >
> > Dear Reviewer,
> >
> > Thank you for your response. We provide some brief clarifications below:
> >
> > > I still think it is rather weak to validate the efficacy of a new optimizer by SFT on Llama.
> >
> > Unfortunately the reviewer may have completely missed our _pre-training results_, presented in the rebuttal. Specifically, in the PDF attached to the main response, we have presented pre-training results for ResNet18 and ResNet50 on ImageNet, trained _from scratch_ showing that our optimizer outperforms Adam and even SGD in this setting, while using _less than half of the memory_ relative to Adam-8bit, and less than 1/4 relative to SGD!
> >
> > Our SFT results on Llama are on 7B and 13B parameter models (the latter is at the end of the rebuttal PDF), where our method presents significant practical advantages--specifically, it allows the user to use a single GPU for SFT, rather than several, while providing a solid convergence theory.
> >
> > We unfortunately do not have the GPU resources currently to perform pre-training at this scale. With all due respect, we do not believe that being able to perform billion-parameter-scale pre-training experiments should be a requirement for acceptance at NeurIPS.
> >
> > > The whole script is written in the style of proposing a new optimizer for generic training procedures (including pre-training), not just for SFT.
> >
> > We respectfully disagree on this point: as the reviewer can see by examining our Limitations section (lines 327 to 331 of the submission), the lack of large-scale pre-training experiments is clearly stated as a limitation of our paper, and extending MicroAdam to large-scale pre-training is cited as our main direction of future work. We believe MicroAdam can be extended to large-scale (LLM) pre-training, but doing so would require significant application-specific effort and major computational resources for validation.
> >
> > We believe we have established the viability of our approach in the submission, both via strong theoretical guarantees and via medium-scale SFT experiments. The rebuttal shows that our approach is also valid for vision pre-training experiments. We would be happy to amend the submission to further clarify the fact that we do not currently aim to do large-scale pre-training, which is left for future work.
> >
> > Respectfully,\
> > The authors

---

### Author Rebuttal · Authors · 2024-08-06

We would like to thank all reviewers for the useful feedback! We have provided individual responses to each review, and briefly summarize the main points here:
- To address the concern about Algorithm 1, we provided a detailed explanation of each line in the algorithm at the end of our response to **Reviewer 8ZAo**.
- To address the concern about the absence of pretraining results, we stated our research question which our work successfully answers (that of reducing the memory usage in the finetuning case), having similar motivation as LoRA [1] (e.g. reducing the memory usage for finetuning, but using a completely different approach). In addition, we also provided results for pretraining Computer Vision tasks, where MicroAdam shows ~2% higher validation accuracy on ResNet-18 and ~5% higher than AdamW variants, while still being 1% better than SGD (see the attached PDF).
- To address the concerns about novelty in our contribution, we emphasized our key algorithmic contributions: compressing the error feedback, providing efficient implementation for GPUs and theoretical guarantees as well.
- To resolve the questions about compressor complexity, compression memory efficiency and computational efficiency, we provided detailed explanations.

We hope our responses address the reviewers’ questions, and would be happy to continue the discussion during the rest of the rebuttal period.

References:
- [1] **LORA: LOW-RANK ADAPTATION OF LARGE LANGUAGE MODELS**, Hu et al., 2021, available at https://arxiv.org/pdf/2106.09685

---

### Comment · Area_Chair_aP5b · 2024-08-11
**Start discussions with authors, please**

Dear Reviewers,

Thank you for your valuable contributions to the review process. As we enter the discussion phase (August 7-13), I kindly request your active participation in addressing the authors' rebuttals and engaging in constructive dialogue.

Please:

- Carefully read the authors' global rebuttal and individual responses to each review.

- Respond to specific questions or points raised by the authors, especially those requiring further clarification from you.

- Engage in open discussions about the paper's strengths, weaknesses, and potential improvements.

- Be prompt in your responses to facilitate a meaningful exchange within the given timeframe.

- Maintain objectivity and professionalism in all communications.

If you have any concerns or need guidance during this process, please don't hesitate to reach out to me.

Your continued engagement is crucial for ensuring a fair and thorough evaluation process.
Thank you for your dedication to maintaining the high standards of NeurIPS.

Best regards,

Area Chair

---

### Decision · Program_Chairs · 2024-09-25

**Decision:**

Accept (poster)

**Comment:**

**Summary:**

MicroAdam is a new optimizer that aims to reduce memory usage compared to Adam while maintaining performance. It uses gradient sparsification, quantization, and error feedback to compress optimizer states. Theoretical convergence guarantees are provided. Experiments show competitive performance on language model fine-tuning tasks with reduced memory usage.

**Strengths:**

- Novel approach to reducing optimizer memory footprint
- Provides theoretical convergence guarantees
- Shows good empirical performance on fine-tuning tasks
- Efficient GPU implementation

**Weaknesses:**

- Initially limited to fine-tuning experiments (though pretraining results were added in rebuttal)
- Theoretical analysis based on AMSGrad rather than Adam
- Complex implementation details for compressors
- Unclear why performance sometimes exceeds uncompressed Adam

**Key Discussion Points:**

- *Pretraining results:* Authors added ImageNet pretraining experiments for ResNet models in rebuttal to address this limitation
- *Theoretical analysis:* Authors clarified choice of AMSGrad as basis due to known issues with Adam's convergence proof
- *Implementation complexity:* Authors acknowledged this but noted existing work/tools can be leveraged
- *Memory savings:* Clarified that main savings are in optimizer state size, which is significantly reduced
- *Performance exceeding Adam:* Authors hypothesized this may be due to regularization effects of compression
- *Applicability beyond fine-tuning:* Additional experiments helped demonstrate broader applicability

Overall, reviewers generally found the work to be technically sound with good theoretical and empirical contributions, though some concerns remained about novelty and broader applicability. The authors' responses and additional experiments addressed many of the initial concerns. Most reviewers maintained or increased their scores to weak accept or higher after the rebuttal.